# CL-DPS: A Contrastive Learning Approach to Blind Nonlinear Inverse Problem Solving via Diffusion Posterior Sampling

**Linfeng Ye[1]\*, Shayan Mohajer Hamidi[2]\*, Mert Pilanci[2], Konstantinos N. Plataniotis[1]**
[1]University of Toronto, [2]Stanford University
[1]`linfeng.ye@mail.utoronto.ca; kostas@ece.utoronto.ca`
[2]`{smohajer,pilanci}@stanford.edu`

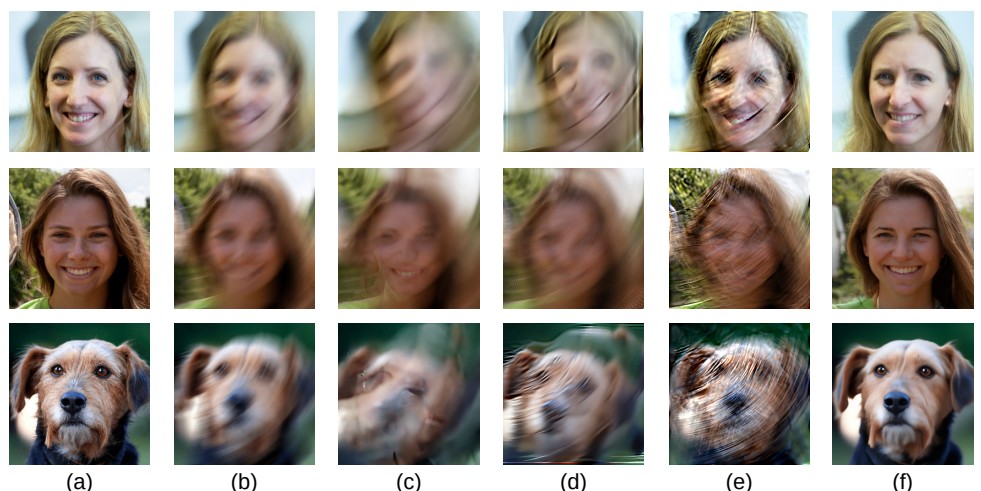

Figure 1: Results of **blind rotational** deblurring, a challenging ***nonlinear*** inverse problem: (a) ground truth, (b) rotation-blurred measurement, and restored images using (c) BlindDPS (Chung et al., 2023a), (d) FastEM (Laroche et al., 2024), (e) GibbsDDRM (Murata et al., 2023), and (f) CL-DPS (ours). All methods fail catastrophically except for CL-DPS.

## ABSTRACT

Diffusion models (DMs) have recently become powerful priors for solving inverse problems. However, most work focuses on non-blind settings with known measurement operators, and existing DM-based blind solvers largely assume *linear* measurements, which limits practical applicability where operators are frequently *nonlinear*. We introduce **CL-DPS**, a **c**ontrastive **l**earning framework for **d**iffusion **p**osterior **s**ampling that requires no knowledge of the operator parameters at inference. To the best of our knowledge, **CL-DPS is the first DM-based framework capable of solving *blind nonlinear* inverse problems**. Our key idea is to train an *auxiliary encoder* offline, using a MoCo-style contrastive objective over randomized measurement operators, to learn a surrogate for the conditional likelihood $p(\boldsymbol{y}|\boldsymbol{x}_t)$. During sampling, we inject the surrogate's gradient as a guidance term along the reverse diffusion trajectory, which enables posterior sampling without estimating or inverting the forward operator. We further employ overlapping patch-wise inference to preserve fine structure and a lightweight color-consistency head to capture color statistics. The guidance is *sampler-agnostic* and pairs well with modern solvers (e.g., *DPM-Solver++ (2M)*). Extensive experiments show that CL-DPS effectively handles challenging *nonlinear* cases, such as rotational and zoom deblurring, where prior DM-based methods fail, while remaining competitive on standard linear benchmarks. Code: `https://anonymous.4open.science/r/CL-DPS-4F5D`.

---

\*Authors contributed equally.

# 1 INTRODUCTION

Inverse problems are pervasive across many fields, with applications in medical imaging (McCann et al., 2017; Jin et al., 2017), computational photography (Tonolini et al., 2020; Ongie et al., 2020), and seismic imaging (Hosseini & Plataniotis, 2020; Zhdanov, 2002). The goal is to recover the original signal $x_0$ from a corrupted measurement $y$ produced by a forward operator $\mathcal{A}_\psi(\cdot)$. Depending on whether this operator is available, inverse problems are categorized as $(i)$ non-blind, where $\mathcal{A}_\psi$ is known, or $(ii)$ blind, where $\mathcal{A}_\psi$ is unknown and must be estimated jointly with $x_0$, making the problem substantially more challenging.

Inverse problems are inherently ill-posed, often relying heavily on data priors $p(x_0)$ for accurate computation. Recently, diffusion models (DMs) have emerged as powerful tools for solving inverse problems due to their remarkable ability to capture complex data distributions $p(x_0)$ (Song et al., 2023; Chung et al., 2023b; Dou & Song, 2024). A straightforward approach to leveraging DMs for solving inverse problems involves training a conditional DM to directly estimate the posterior $p(x_0|y)$ via supervised learning. However, this method can be computationally intensive, as it requires training separate DMs for each distinct measurement operator $\mathcal{A}_\psi$.

To overcome this limitation, a practical alternative uses a pretrained, unconditional DM as a prior for $p(x_0)$ and combines it with a likelihood term inside a diffusion posterior sampling (DPS) scheme. During sampling, the DM maintains a latent variable $x_t$ that represents the current state along the reverse diffusion process at step $t$. Posterior guidance needs a surrogate for the intractable likelihood $p(y|x_t)$, which recent work approximates in various ways (Chung et al., 2023b; Song et al., 2023).

Nevertheless, most DM-based inverse solvers remain limited to non-blind settings where the measurement operator $\mathcal{A}_\psi$ is assumed known (Chung et al., 2023b; Song et al., 2023). However, accurately obtaining the measurement operator is often difficult or infeasible in real-world applications (Chung et al., 2023a; Laroche et al., 2024; Ji et al., 2024). Recently, efforts have emerged in the literature to address blind inverse problems using DMs. In particular, Chung et al. (2023a) introduced BlindDPS, a method that trains a DM specifically for the blur operator. While this approach benefits from widely available pretrained DMs for signals such as images and audio, it requires additional training of a DM for the parameters of the relevant linear operators, significantly limiting its practical applicability. Murata et al. (2023) proposed GibbsDDRM, which constructs a joint distribution over the data, measurements, and linear operator, addressing the problem through posterior sampling using a variant of the Gibbs sampler. Similarly, Sanghvi (2024) estimated the kernel parameters to tackle deconvolution problems. However, these methods are fundamentally restricted to addressing *only* linear inverse problems, as they **assume that $\mathcal{A}_\psi$ is a convolution operator** (Chihaoui et al., 2024; Sanghvi, 2024; Murata et al., 2023; Chung et al., 2023a). In practice, however, many blind inverse problems involve operators that are nonlinear, rendering these approaches inadequate for such cases.

To address the aforementioned limitation, we propose **CL-DPS**, a method based on **c**ontrastive **l**earning for solving blind inverse problems via **d**iffusion **p**osterior **s**ampling. To the best of our knowledge, CL-DPS is the ***first*** DM-based method that can solve *blind nonlinear* inverse problems, without requiring knowledge or estimation of the operator parameters. Concretely, in CL-DPS, an auxiliary encoder is first trained ***offline*** using a modified version of MoCo (He et al., 2020), a contrastive learning (CL) technique. The role of this auxiliary encoder is to estimate the likelihood $p(y|x_t)$ without knowing the measurement $\mathcal{A}_\psi$. Then, during inverse problem solving, we perform inference with this auxiliary encoder to estimate $p(y|x_t)$, which is then used to guide the reverse path of the diffusion process. To further improve the auxiliary encoder's accuracy in estimating $p(y|x_t)$, we introduce a novel overlapping patch-wise inference method that divides the images into patches during the inference stage.

We evaluate CL-DPS on FFHQ (Karras et al., 2021), AFHQ (Choi et al., 2020), and ImageNet (Deng et al., 2009) under blind linear and nonlinear measurements. In the *nonlinear* setting (e.g., rotation blur), prior DM-based methods fail, while CL-DPS restores high-quality images (see Figure 1). In the *linear* setting, CL-DPS is competitive with state of the art. Our main contributions are:

• We introduce CL-DPS, a diffusion posterior sampling framework that learns a contrastive likelihood surrogate offline and plugs it in as guidance at test time. The auxiliary encoder is trained with a MoCo-style framework over randomized measurements, so CL-DPS applies to both linear and

nonlinear blind settings without knowing the measurement parameters at inference. In addition, a lightweight color-consistency head is deployed to capture color information.

• We establish a lemma (Lemma 1) showing that, under an energy-based formulation, the gradient of the contrastive log probability coincides with the desired likelihood gradient and becomes consistent as the dictionary size grows. We further present a denominator-aware variant, which yields results comparable to the simpler numerator-only update but incurs higher computational cost.

• We introduce an overlapping patch-wise inference strategy with an information-theoretic guarantee (Theorem 1): stacking more overlapping patch features increases the mutual information between the signal and its encoded features, $\text{I}(\boldsymbol{x}; f(\{\boldsymbol{p}_j^{\boldsymbol{x}}\}))$.

• Across blind linear and nonlinear tasks, including rotation blur, CL-DPS delivers strong quantitative and qualitative results. Ablations isolate the impact of patch-wise inference, color regularization, and dictionary size, and confirm the efficiency of the numerator-only guidance.

## 2 RELATED WORK AND NOTATION

• **Diffusion models for inverse problems:** The use of DMs to solve inverse problems through posterior sampling has recently attracted considerable attention across various domains, including image denoising (Kawar et al., 2022; Hamidi & Yang, 2025b;a), compressed sensing (Bora et al., 2017; Kadkhodaie & Simoncelli, 2021), magnetic resonance imaging (MRI) (Jalal et al., 2021), score-based stochastic differential equations (SDEs) (Song et al., 2022), and variational methods (Mardani et al.; Feng & Bouman, 2023). For non-blind inverse problems, methods such as diffusion posterior sampling (DPS) (Chung et al., 2023b) and pseudo-guided diffusion models (ΠGDM) (Song et al., 2023) leverage Tweedie's formula (Efron, 2011) to approximate the smoothed likelihood. Similarly, singular-value decomposition based techniques (Kawar et al., 2021) are applied for related purposes. In addition, Hamidi et al. (2025) eliminates the need for constraint tuning or likelihood approximation by coupling data and measurement space diffusion posterior sampling.

On the other hand, for blind inverse problems, alongside the approaches discussed in Section 1 (Chung et al., 2023a; Murata et al., 2023; Sanghvi, 2024), Alkan et al. (2023) introduced Blind RED-Diff, an extension of the RED-Diff framework (Mardani et al.). This method employs variational inference to jointly estimate both the latent image and the unknown forward model parameters, addressing the challenges of unknown measurement operators. Recent work applies diffusion priors to amortized variational inference for inverse problems (Lee et al., 2024), training a network end to end to map $\boldsymbol{y}$ to posterior parameters under a specified forward model. Our setting is different: *blind nonlinear* operators (no operator estimates). We keep the diffusion prior frozen and use a contrastive plug in likelihood surrogate to guide sampling from $(\boldsymbol{x}_t, \boldsymbol{y})$.

We defer the discussion of contrastive learning to Appendix A.

• **Notation:** For a positive integer $C$, let $[C] \triangleq \{1, \ldots, C\}$. Scalars are denoted by non-bold letters (e.g. $u$ and $U$), vectors by boldface lowercase letters (e.g. $\boldsymbol{u}$). Denote by $\boldsymbol{u}[i]$ the $i$-th element of vector $\boldsymbol{u}$. For two vectors $\boldsymbol{u}$ and $\boldsymbol{v}$, denote by $\langle \boldsymbol{u}, \boldsymbol{v} \rangle$ their inner product. We use $|\mathcal{C}|$ to denote the cardinality of a set $\mathcal{C}$. $(\cdot)^\mathsf{T}$ denotes the transpose operation. The mutual information between two random variables $X$ and $Y$ is denoted by $\text{I}(X; Y)$.

## 3 BACKGROUND AND PRELIMINARIES

### 3.1 DIFFUSION MODELS

DMs generate data by reversing a forward noising process. We adopt the variance preserving SDE (VP-SDE) (Song et al., 2020), which is equivalent to DDPM (Ho et al., 2020)

$$d\boldsymbol{x} \ = \ -\tfrac{\beta_t}{2}\,\boldsymbol{x}\,dt \ + \ \sqrt{\beta_t}\,d\boldsymbol{w}, \tag{1}$$

with noise schedule $\beta_t > 0$ and standard Wiener process $\boldsymbol{w}$. The data distribution is at $t = 0$ with $\boldsymbol{x}_0 \sim p_{\text{data}}$ and at $t = T$ the state is $\boldsymbol{x}_T \sim \mathcal{N}(\boldsymbol{0}, \boldsymbol{I})$.

The reverse time SDE (Anderson, 1982) is

$$dx = \left[ -\frac{\beta_t}{2} x - \beta_t \nabla_{x_t} \log p(x_t) \right] dt + \sqrt{\beta_t} d\bar{w}, \tag{2}$$

where $dt$ flows backward and $d\bar{w}$ is reverse time Wiener noise. The score $\nabla_{x_t} \log p(x_t)$ is approximated by a neural network $s_\theta$ trained with denoising score matching (Vincent, 2011)

$$\theta^* = \arg\min_\theta \mathbb{E}_{t \sim U(\varepsilon, 1)} \mathbb{E}_{x_0 \sim p_{\text{data}}} \mathbb{E}_{x_t \sim p(x_t | x_0)} \left[ \| s_\theta(x_t, t) - \nabla_{x_t} \log p(x_t \mid x_0) \|_2^2 \right]. \tag{3}$$

After training we use $s_{\theta^*}(x_t, t)$ as an estimate of the score in Equation (2). Discretizing Equation (2) yields the DDPM sampler. We use $\alpha_i \triangleq 1 - \beta_i$ and $\bar{\alpha}_i \triangleq \prod_{j=1}^{i} \alpha_j$ for the discrete schedule.

### 3.2 DIFFUSION MODELS FOR SOLVING INVERSE PROBLEMS

We observe $y \in \mathbb{R}^m$ from an unknown $x_0 \in \mathbb{R}^d$ via

$$y = \mathcal{A}_\psi(x_0) + n, \tag{4}$$

where $\mathcal{A}_\psi$ is a linear or nonlinear measurement with unknown parameters $\psi$ and $n$ is Gaussian with zero-mean and covariance $\sigma^2 I$. The regime $m < d$ is ill-posed and requires a prior on $x_0$.

Using a diffusion prior we sample from the posterior by modifying the reverse SDE to include the likelihood term

$$dx = \left[ -\frac{\beta_t}{2} x - \beta_t \big( \nabla_{x_t} \log p(x_t) + \nabla_{x_t} \log p(y \mid x_t) \big) \right] dt + \sqrt{\beta_t} d\bar{w}, \tag{5}$$

which follows from

$$\nabla_{x_t} \log p(x_t \mid y) = \nabla_{x_t} \log p(x_t) + \nabla_{x_t} \log p(y \mid x_t). \tag{6}$$

The prior score $\nabla_{x_t} \log p(x_t)$ is given by the pretrained network $s_{\theta^*}$. The bottleneck is the likelihood term $\nabla_{x_t} \log p(y \mid x_t)$ which is time dependent and intractable when $\psi$ is unknown.

Prior work often assumes a known operator for likelihood evaluation (Chung et al., 2022a; 2023b). We address the blind case by learning a contrastive likelihood surrogate offline and using it as guidance inside the sampler, as detailed in Section 4.

### 3.3 MOMENTUM CONTRAST LEARNING

Contrastive learning learns representations by pulling together positives and pushing apart negatives (Hadsell et al., 2006; Tian et al., 2020). MoCo (He et al., 2020) implements this as dictionary lookup with two components: a large *queue* of keys that serves as a dynamic dictionary and a *momentum encoder* that produces stable keys. Let $f_q$ be the query encoder with parameters $\theta_q$ and $f_k$ the key encoder with parameters $\theta_k$ updated as $\theta_k \leftarrow m\theta_k + (1-m)\theta_q$, where $m \in [0, 1)$ is the momentum coefficient. Given a query $q = f_q(\cdot)$ a positive key $k_+ = f_k(\cdot)$ and $K$ negative keys $\{k_i\}_{i=1}^{K}$ from the queue MoCo minimizes the InfoNCE loss (Oord et al., 2018)

$$\mathcal{L}_q = -\log \frac{\exp\big(\langle q, k_+ \rangle / \tau\big)}{\sum_{i=1}^{K} \exp\big(\langle q, k_i \rangle / \tau\big)}, \tag{7}$$

with temperature $\tau > 0$. In our method we set $q = f(x_t)$ the positive key to $k_+ = f(y_{\text{syn}})$ and use the MoCo queue to approximate the dictionary $Y$ of negatives.

## 4 METHODOLOGY

As discussed in Section 3, diffusion posterior sampling relies on the likelihood term $p(y \mid x_t)$ and its gradient with respect to $x_t$. We approximate this term with an auxiliary encoder $f$, trained **offline** to provide a surrogate across a range of measurement operators since the parameters $\psi$ are unknown at inference. During reverse diffusion, $f$ supplies the likelihood gradient to guide the sampling trajectory. Section 4.1 details the contrastive learning procedure used to train $f$, Section 4.2 shows how it is applied at inference, and Section 4.3 presents the full CL-DPS algorithm.

### 4.1 TRAINING THE AUXILIARY ENCODER

#### 4.1.1 CONTRASTIVE LEARNING AS LIKELIHOOD ESTIMATION

Using Bayes' formula, the likelihood $p(\boldsymbol{y}|\boldsymbol{x}_t)$ can be written as

$$p(\boldsymbol{y} \mid \boldsymbol{x}_t) = \frac{p(\boldsymbol{y}, \boldsymbol{x}_t)}{p(\boldsymbol{x}_t)} = \frac{p(\boldsymbol{y}, \boldsymbol{x}_t)}{\int p(\tilde{\boldsymbol{y}}, \boldsymbol{x}_t) \, d\tilde{\boldsymbol{y}}}. \tag{8}$$

To compute Equation (8), we first obtain a numerical representation of its numerator, $p(\boldsymbol{y}, \boldsymbol{x}_t)$. Specifically, following (Oord et al., 2018; Li et al., 2021), we approximate it with an energy score learned by a neural encoder $f$ as: $p(\boldsymbol{y}, \boldsymbol{x}_t) \propto \exp\big(\langle f(\boldsymbol{x}_t), f(\boldsymbol{y})\rangle / \tau\big)$, where $\tau > 0$ is a temperature.

The denominator in Equation (8), $\int p(\tilde{\boldsymbol{y}}, \boldsymbol{x}_t) d\tilde{\boldsymbol{y}}$, is generally intractable. Thus, we rely on an approximation method, using a finite sum as follows: $\int p(\tilde{\boldsymbol{y}}, \boldsymbol{x}_t) d\tilde{\boldsymbol{y}} \approx \sum_{\tilde{\boldsymbol{y}} \in Y} p(\tilde{\boldsymbol{y}}, \boldsymbol{x}_t)$, where $Y$ is a sufficiently large set. This allows us to numerically approximate $p(\boldsymbol{y} \mid \boldsymbol{x}_t)$ as:

$$p(\boldsymbol{y} \mid \boldsymbol{x}_t) \approx \frac{\exp(\langle f(\boldsymbol{x}_t), f(\boldsymbol{y})\rangle/\tau)}{\sum_{\tilde{\boldsymbol{y}} \in Y} \exp(\langle f(\boldsymbol{x}_t), f(\tilde{\boldsymbol{y}})\rangle/\tau)}. \tag{9}$$

This suggests training $f$ by maximizing the log of Equation (9). Equivalently we minimize the negative log likelihood surrogate

$$\mathcal{L}_{p(\boldsymbol{y}|\boldsymbol{x}_t)} = -\log \frac{\exp(\langle f(\boldsymbol{x}_t), f(\boldsymbol{y})\rangle/\tau)}{\sum_{\tilde{\boldsymbol{y}} \in Y} \exp(\langle f(\boldsymbol{x}_t), f(\tilde{\boldsymbol{y}})\rangle/\tau)}. \tag{10}$$

Comparing Equation (10) with the InfoNCE loss in Equation (7) shows that the standard contrastive objective is a direct estimator of this surrogate when the query is $q = f(\boldsymbol{x}_t)$ and the keys are $\{k_i\}_{i \in [K]} \subset Y$. With a large dictionary as in MoCo, the queue of size $K$ provides a practical approximation to the population $Y$.

To justify that this contrastive surrogate serves as the likelihood term required by diffusion posterior sampling, we state the following Lemma that links the contrastive log probability to the conditional likelihood gradient. The full version with assumptions and the proof are provided in Appendix B.

**Lemma 1** (Contrastive likelihood gradient, *short version*)**.** *Let $\tau > 0$ and define*

$$s_t(\boldsymbol{x}_t \mid \boldsymbol{y}) \triangleq \langle f(\boldsymbol{x}_t), f(\boldsymbol{y})\rangle / \tau. \tag{11}$$

*For a finite dictionary $Y$ with $\boldsymbol{y} \in Y$ define the softmax surrogate*

$$\widehat{p}_{t,Y}(\boldsymbol{y} \mid \boldsymbol{x}_t) = \frac{\exp\{s_t(\boldsymbol{x}_t \mid \boldsymbol{y})\}}{\sum_{\tilde{\boldsymbol{y}} \in Y} \exp\{s_t(\boldsymbol{x}_t \mid \tilde{\boldsymbol{y}})\}}. \tag{12}$$

*Then,*

$$\nabla_{\boldsymbol{x}_t} \log \widehat{p}_{t,Y}(\boldsymbol{y} \mid \boldsymbol{x}_t) = \nabla_{\boldsymbol{x}_t} s_t(\boldsymbol{x}_t \mid \boldsymbol{y}) - \sum_{\tilde{\boldsymbol{y}} \in Y} \widehat{p}_{t,Y}(\tilde{\boldsymbol{y}} \mid \boldsymbol{x}_t) \nabla_{\boldsymbol{x}_t} s_t(\boldsymbol{x}_t \mid \tilde{\boldsymbol{y}}), \tag{13}$$

*Moreover, under the energy model $p(\boldsymbol{y} \mid \boldsymbol{x}_t) \propto \exp\{s_t(\boldsymbol{x}_t \mid \boldsymbol{y})\}$ with mild integrability conditions stated in Appendix B and with $Y_n$ drawn i.i.d. and augmented so that $\boldsymbol{y} \in Y_n$*

$$\nabla_{\boldsymbol{x}_t} \log \widehat{p}_{t,Y_n}(\boldsymbol{y} \mid \boldsymbol{x}_t) \xrightarrow[n \to \infty]{a.s.} \nabla_{\boldsymbol{x}_t} \log p(\boldsymbol{y} \mid \boldsymbol{x}_t). \tag{14}$$

#### 4.1.2 TRAINING THE AUXILIARY ENCODER FOR DPS

To optimize Equation (10) we ideally need pairs $(\boldsymbol{x}_t, \boldsymbol{y})$ drawn from the true measurement process since inference will condition on the observed $\boldsymbol{y}$. During training the measurement parameters $\psi$ of the operator family $\mathcal{A}_\psi$ are unknown and vary at test time, so we replace $\boldsymbol{y}$ by a surrogate synthetic measurement $\boldsymbol{y}_{\text{syn}}$ generated by sampling $\psi$ from a prior $P_\Psi$. This trains the encoder to approximate the likelihood across a range of operator parameter settings. To this end, given a clean image $\boldsymbol{x}_0$ we synthesize the training pair $(\boldsymbol{x}_t, \boldsymbol{y}_{\text{syn}})$ at a randomly chosen time $t$ as

$$\boldsymbol{y}_{\text{syn}} = \mathcal{A}_\psi(\boldsymbol{x}_0), \qquad \boldsymbol{x}_t = \sqrt{\bar{\alpha}_t}\, \boldsymbol{x}_0 + \sqrt{1 - \bar{\alpha}_t}\, \boldsymbol{n}, \tag{15}$$

where $\psi \sim P_\Psi$ and $n \sim \mathcal{N}(\mathbf{0}, \mathbf{I})$. The diffusion schedule satisfies $\bar{\alpha}_t = \prod_{j=1}^{t} \alpha_j$ with $0 < \alpha_j < 1$. Examples of $\mathcal{A}_\psi$ include Gaussian, motion, rotation, and zoom blur, where the operator parameters $\psi$ are sampled from the prior $P_\Psi$, which is commonly chosen as a uniform distribution over a specified range of blur intensities. At each training iteration, we construct a distinct pair $(\boldsymbol{x}_t, \boldsymbol{y}_{\text{syn}})$. Note that although $f$ does not take $t$ explicitly, its input $\boldsymbol{x}_t$ is drawn at timestep $t$, so the surrogate $p(\boldsymbol{y} \,|\, \boldsymbol{x}_t)$ (and its gradient) are implicitly $t$-dependent.

The contrastive dictionary $Y$ is formed by the current mini-batch's embeddings (EBDs) together with a queue of EBDs from previous batches; the EBDs are produced by a momentum-updated encoder, as in MoCo. We train the encoder with an InfoNCE-style objective $\mathcal{L}_{p(\boldsymbol{y}_{\text{syn}}|\boldsymbol{x}_t)}$.

In practice, the contrastive objective can become insensitive to color information, often resulting in hue or brightness shifts in reconstructions. To mitigate this issue, we add a lightweight color-consistency head (CCH), denoted by $\mathsf{H}_c$, on top of the auxiliary encoder. The CCH is trained to predict the global color statistics of the input. Formally, let $\boldsymbol{x}_t \in \mathbb{R}^{C \times N_1 \times N_2}$ and define its spatial average

$$\left[\mathsf{AP}(\boldsymbol{x}_t)\right]_c = \frac{1}{N_1 N_2} \sum_{i=1}^{N_1} \sum_{j=1}^{N_2} \boldsymbol{x}_{t\,cij}.$$

The color-consistency head $\mathsf{H}_c(\boldsymbol{x}_t) \in \mathbb{R}^C$ is implemented as a two-layer convolutional module with global pooling followed by a sigmoid activation. We define the color-consistency loss as

$$\mathcal{L}_{CC}(\boldsymbol{x}_t) = \left\| \mathsf{H}_c(\boldsymbol{x}_t) - \mathsf{AP}(\boldsymbol{x}_t) \right\|_2^2.$$

To train the auxiliary encoder $f$ to approximate the likelihood and capture color information, we optimize the following combined loss:

Figure 2: Overview of the training process for the auxiliary encoder. The figure also illustrates the structure of the linear projection head and the color-consistency head (CCH). The CCH is a two-layer convolutional network that encourages the model to preserve the color information of the input during training. 🔥 indicates trainable components, sg stands for stop gradient, and ema for exponential moving average.

$$\mathcal{L}_{\text{CL-DPS}} = \mathcal{L}_{p(\boldsymbol{y}_{\text{syn}}|\boldsymbol{x}_t)} + \lambda \mathcal{L}_{CC}(\boldsymbol{x}_t), \tag{16}$$

where $\lambda > 0$ balances likelihood estimation and color preservation. The loss in Equation (16) is averaged over the mini-batch. The CCH is used only during training and discarded at inference. At test time, $\boldsymbol{y}_{\text{syn}}$ is replaced by the observed measurement $\boldsymbol{y}$, and the pretrained encoder $f$ provides surrogate likelihood guidance within CL-DPS. The framework for CL-DPS is depicted in Figure 2.

**Remark 1.** *CL-DPS adds a small, one time auxiliary training stage to enable the blind nonlinear setting. This stage is lightweight, and the encoder is trained once for a given forward setting and then reused for all measurements and noise realizations. The procedure remains zero shot for the downstream task because it uses only synthetic pairs generated from the diffusion prior and the measurement simulator, no extra labels or human annotations are needed. In practice this offline cost is far smaller than pretraining or fine tuning a DM or training a supervised reconstructor, and per image inference remains on the same order as standard DPS (see Appendix K). The added training enables, to our knowledge, the first DM-based solution to blind nonlinear inverse problems without estimating operator parameters, which prior training free methods do not handle.*

### 4.2 LIKELIHOOD ESTIMATION USING AUXILIARY ENCODER

After training with Equation (16), the encoder $f$ provides a surrogate for $p(\boldsymbol{y}|\boldsymbol{x}_t)$ that we use inside diffusion posterior sampling. However, the convolutional encoders often compress low level details (Yang et al., 2025; Chi et al., 2025; Tishby & Zaslavsky, 2015; Ye et al., 2024; Yang & Ye, 2024),

which can reduce the granularity needed for inverse problems. We therefore use an overlapping patch-wise **inference** scheme that increases how much information the encoder output retains about its input.

### 4.2.1 Overlapping Patch-Wise Inference

Given an image $\boldsymbol{x} \in \mathbb{R}^{N_1 \times N_2}$ we partition it into $L_s$ overlapping $n \times n$ patches with stride $s < n$ denoted by $\{\boldsymbol{p}_j^{\boldsymbol{x}}\}_{j \in [L_s]}$, where $L_s = \left\lfloor \frac{N_1 - n}{s} + 1 \right\rfloor \left\lfloor \frac{N_2 - n}{s} + 1 \right\rfloor$. Then, we run $f$ on each patch and stack the features

$$f\big(\{\boldsymbol{p}_j^{\boldsymbol{x}}\}_{j \in [L_s]}\big) = \big[\, f^{\mathsf{T}}(\boldsymbol{p}_1^{\boldsymbol{x}}) \, \dots \, f^{\mathsf{T}}(\boldsymbol{p}_{L_s}^{\boldsymbol{x}}) \,\big]^{\mathsf{T}}. \tag{17}$$

To quantify retained information we use mutual information $\mathrm{I}(\boldsymbol{x}; f(\boldsymbol{x}))$. The next result shows that stacking more overlapping patch features increases this quantity.

**Theorem 1.** *Let $\boldsymbol{x} \in \mathbb{R}^{N_1 \times N_2}$ be any random image. Fix a patch size $n \times n$ and a stride $s < n$. Let $f : \mathbb{R}^{n \times n} \to \mathbb{R}^d$ act patch-wise. For integers $1 \leq U < V$ extract $U$ and $V$ overlapping patches and form stacked features as above. Then,*

$$\mathrm{I}\big(\boldsymbol{x}; f(\{\boldsymbol{p}_j^{\boldsymbol{x}}\}_{j \in [U]})\big) \leq \mathrm{I}\big(\boldsymbol{x}; f(\{\boldsymbol{p}_j^{\boldsymbol{x}}\}_{j \in [V]})\big). \tag{18}$$

The proof is deferred to Appendix C. In words, denser overlapping patching makes the encoder output more informative about $\boldsymbol{x}$.

**Corollary 1.** *Adopt the notation of Theorem 1. For a single extracted patch ($U = 1$), write $f(\boldsymbol{x}) \triangleq f\big(\{\boldsymbol{p}_1^{\boldsymbol{x}}\}\big) \in \mathbb{R}^d$. Then for every integer $U \geq 1$, we have $\mathrm{I}\left(\boldsymbol{x}; f(\boldsymbol{x})\right) \leq \mathrm{I}\left(\boldsymbol{x}; f(\{\boldsymbol{p}_j^{\boldsymbol{x}}\}_{j \in [U]})\right)$.*

We further complement Theorem 1 and Corollary 1 with a variance-reduction analysis for the guidance estimator, simple design rules, and an empirical study in Appendix D.

During likelihood estimation we apply the same patchification to the measurement $\boldsymbol{y}$ and use the stacked features to score consistency.

**Remark 2.** *Patch-wise ideas exist in prior work. Hu et al. (2024); Zhang et al. (2025) train a patch-based diffusion prior and aggregate patch scores with positional encoding to form a whole-image prior score. Wang et al. propose DDNM, a* zero-shot *restoration method that uses an off-the-shelf diffusion prior and enforces data consistency by modifying only the null-space component for linear measurements, without auxiliary training. In contrast, our overlapping patch-wise step is* neither *a prior* nor *a null-space projector. It extracts stacked local features that are fed to an auxiliary encoder learning a contrastive likelihood surrogate for DPS guidance.*

---

**Algorithm 1: CL-DPS (ours)**

1: **Input** number of steps $T$, measurement $\boldsymbol{y}$, noise schedule $\{\tilde{\sigma}_t\}$, *pretrained* encoder $f(\cdot)$, step size $\eta > 0$, number of overlapping patches $U$.
2: $\boldsymbol{x}_T \sim \mathcal{N}(\mathbf{0}, \boldsymbol{I})$.     // initialize with Gaussian noise
3: Extract $U$ overlapping patches from $\boldsymbol{y}$ once.
    // cache measurement features
4: $\{\boldsymbol{p}_j^{\boldsymbol{y}}\}_{j \in [U]} \leftarrow \boldsymbol{y}$.
5: **for** $t = T-1 \dots 0$ **do**
6:     $\hat{\boldsymbol{s}} \leftarrow \boldsymbol{s}_\theta(\boldsymbol{x}_t, t)$.     // score model estimate of $\nabla_{\boldsymbol{x}_t} \log p(\boldsymbol{x}_t)$
7:     $\tilde{\boldsymbol{x}}_0 \leftarrow \frac{1}{\sqrt{\bar{\alpha}_t}}\big(\boldsymbol{x}_t + (1 - \bar{\alpha}_t)\hat{\boldsymbol{s}}\big)$.     // Tweedie posterior mean
8:     $\boldsymbol{z} \sim \mathcal{N}(\mathbf{0}, \boldsymbol{I})$.
9:     $\boldsymbol{x}_{t-1}' \leftarrow \frac{\sqrt{\alpha_t}(1 - \bar{\alpha}_{t-1})}{1 - \bar{\alpha}_t} \boldsymbol{x}_t + \frac{\sqrt{\bar{\alpha}_{t-1}}\beta_t}{1 - \bar{\alpha}_t} \tilde{\boldsymbol{x}}_0 + \tilde{\sigma}_t \boldsymbol{z}$.
    // DDPM update
10:    Extract $U$ overlapping patches from $\boldsymbol{x}_t$.
11:    $\{\boldsymbol{p}_j^{\boldsymbol{x}_t}\}_{j \in [U]} \leftarrow \boldsymbol{x}_t$.
12:    $\boldsymbol{x}_{t-1} \leftarrow \boldsymbol{x}_{t-1}' - \eta \nabla_{\boldsymbol{x}_t} \big\langle f(\{\boldsymbol{p}_j^{\boldsymbol{x}_t}\}_{j \in [U]}), \, f(\{\boldsymbol{p}_j^{\boldsymbol{y}}\}_{j \in [U]}) \big\rangle$.
    // contrastive guidance
13: **end for**
14: **Output** $\boldsymbol{x}_0$.

---

### 4.3 Algorithm for CL-DPS

After training with Equation (16), we keep only the encoder $f$ and use it as a likelihood surrogate inside diffusion posterior sampling. We integrate $f$ into DPS (Chung et al., 2023b) with overlapping patch-wise features from Section 4.2. The only change relative to unconditional sampling is a contrastive guidance step that adds an estimate of $\nabla_{\boldsymbol{x}_t} \log p(\boldsymbol{y} \mid \boldsymbol{x}_t)$ at each time $t$. The procedure is summarized in Algorithm 1. In line 12 of Algorithm 1, we adopt an energy-guidance view: treat the contrastive score as an unnormalized likelihood and use only the numerator gradient, following prior

Table 1: **Nonlinear** blind inverse problems: Blind rotation and zoom deblurring results on the FFHQ, AFHQ and ImageNet datasets. Only CL-DPS achieves high-quality image restoration; other methods fail. **Bold** and underlined values denote the best and second-best results, respectively.

| | **Rotation** | | | | | | | | |
|---|---|---|---|---|---|---|---|---|---|
| | FFHQ ($256 \times 256$) | | | AFHQ ($256 \times 256$) | | | ImageNet | | |
| Method | PSNR ↑ | FID ↓ | LPIPS ↓ | PSNR ↑ | FID ↓ | LPIPS ↓ | PSNR ↑ | FID ↓ | LPIPS ↓ |
| CL-DPS (SPE) | **22.74** | **33.66** | **0.302** | **21.46** | **36.96** | **0.319** | **20.05** | **45.10** | **0.340** |
| CL-DPS (UNI) | 22.27 | 36.55 | 0.315 | 21.61 | 39.81 | 0.330 | 19.92 | 49.23 | 0.352 |
| FastEM WACV 2024 | 15.96 | 268.4 | 0.597 | 11.57 | 289.2 | 0.684 | 13.90 | 337.8 | 0.721 |
| BlindDPS CVPR 2023a | 16.87 | 343.8 | 0.552 | 13.25 | 200.5 | 0.674 | 11.25 | 392.4 | 0.895 |
| GibbsDDRM ICML 2023 | 18.43 | 236.6 | 0.565 | 15.24 | 263.5 | 0.628 | 12.24 | 311.2 | 0.781 |
| Pan-$\ell_0$ TPAMI 2017 | 14.63 | 327.7 | 0.629 | 13.41 | 227.8 | 0.895 | 11.52 | 340.7 | 0.862 |
| | **Zoom** | | | | | | | | |
| | FFHQ ($256 \times 256$) | | | AFHQ ($256 \times 256$) | | | ImageNet | | |
| Method | PSNR ↑ | FID ↓ | LPIPS ↓ | PSNR ↑ | FID ↓ | LPIPS ↓ | PSNR ↑ | FID ↓ | LPIPS ↓ |
| CL-DPS (SPE) | **20.68** | **42.61** | **0.435** | **19.63** | **57.54** | **0.468** | **18.56** | **55.30** | **0.481** |
| CL-DPS (UNI) | 20.31 | 46.83 | 0.448 | 19.23 | 61.06 | 0.480 | 18.07 | 59.53 | 0.492 |
| FastEM WACV 2024 | 17.68 | 303.4 | 0.623 | 15.69 | 310.1 | 0.797 | 12.76 | 331.4 | 0.754 |
| BlindDPS CVPR 2023a | 16.39 | 292.9 | 0.784 | 11.75 | 279.6 | 0.607 | 11.96 | 348.2 | 0.812 |
| GibbsDDRM ICML 2023 | 15.45 | 327.4 | 0.802 | 14.57 | 280.5 | 0.549 | 11.14 | 299.8 | 0.702 |
| Pan-$\ell_0$ TPAMI 2017 | 11.52 | 392.2 | 0.715 | 12.41 | 292.8 | 0.851 | 9.452 | 347.2 | 0.743 |

unnormalized-energy guidance (Lu et al., 2023; Du et al., 2023). A denominator-aware alternative is provided in Appendix E, where we show that it yields essentially the same reconstruction quality but with higher computational cost.

**Remark 3.** *We note that CL-DPS can be seamlessly combined with alternative diffusion sampling processes. For example, in Appendix F, we pair CL-DPS with* DPM-Solver++ (2M) *(Lu et al., 2025), providing the algorithm, implementation details, and representative results.*

## 5 EXPERIMENTS

### 5.1 EXPERIMENTAL SETUP

• **Implementation details.** Implementation details for CL-DPS, including training the auxiliary encoder, and implementation details for baseline methods are provided in Appendix G.

**Datasets.** We use Flickr-faces-HQ (FFHQ) $256 \times 256$ dataset (Karras et al., 2021) and animal faces-HQ (AFHQ) $256 \times 256$ dataset (Choi et al., 2020) and ImageNet (Deng et al., 2009).

• **Pretrained diffusion models.** We leverage pretrained score functions from Chung et al. (2022b).

• **Evaluation metrics.** We use Fréchet inception distance (FID) (Heusel et al., 2017), learned perceptual image patch similarity (LPIPS) (Zhang et al., 2018) and peak signal-to-noise ratio (PSNR) between the original image and the reconstructed image as the evaluation metrics.

• **Choice of benchmarks.** For the *blind nonlinear* setting, to the best of our knowledge there are no prior DM-based methods; we therefore compare against the strongest *blind linear* DM-based solvers (FastEM (Laroche et al., 2024), BlindDPS (Chung et al., 2023a), GibbsDDRM (Murata et al., 2023)) and include one classical non-DM baseline, Pan-$\ell_0$ (Pan et al., 2017). For the *blind linear* setting, we evaluate against the full set of seven baselines: SelfDeblur (Ren et al., 2020), DeblurGANv2 (Kupyn et al., 2019), Pan-$\ell_0$ (Pan et al., 2017), BlindDPS (Chung et al., 2023a), FastEM (Laroche et al., 2024), LatentDEM (Bai et al., 2025), and GibbsDDRM (Murata et al., 2023). The last four methods use DMs. CL-DPS is not the only method that uses synthetic degradations for training. In particular, BlindDPS and FastEM train operator priors on synthetic blur kernels, while DeblurGANv2 is trained on synthetic degraded–clean image pairs.

For CL-DPS, we evaluate two training regimes: (*i*) *Universal* (CL-DPS (UNI)), using a single encoder trained jointly across all operator families; and (*ii*) *Specialist* (CL-DPS (SPE)), training a separate encoder offline for each family $\mathcal{A}_\psi$ (e.g., distinct encoders for rotation and zoom blur). At inference, we select the encoder for the detected family while still treating $\psi$ as unknown. See Appendix H for justification.

Table 2: **Linear** blind inverse problems: blind motion and Gaussian deblurring results. **Bold** and underlined values denote the best and second-best results, respectively.

| | **Motion** | | | | | | | | |
|---|---|---|---|---|---|---|---|---|---|
| | FFHQ ($256 \times 256$) | | | AFHQ ($256 \times 256$) | | | ImageNet | | |
| Method | PSNR↑ | FID↓ | LPIPS↓ | PSNR↑ | FID↓ | LPIPS↓ | PSNR↑ | FID↓ | LPIPS↓ |
| CL-DPS (SPE) | **26.33** | **27.44** | 0.117 | **24.06** | 26.25 | **0.186** | **22.27** | 40.35 | **0.131** |
| CL-DPS (UNI) | 26.17 | 28.90 | 0.124 | 23.72 | 28.90 | 0.191 | 21.88 | 43.20 | 0.135 |
| SelfDeblur CVPR 2020 | 10.83 | 270.0 | 0.717 | 9.082 | 300.5 | 0.768 | 9.542 | 320.1 | 0.775 |
| DeblurGANv2 ICCV 2019 | 17.75 | 220.7 | 0.571 | 17.64 | 186.2 | 0.597 | 18.40 | 260.2 | 0.561 |
| Pan-$\ell_0$ TPAMI 2017 | 15.53 | 242.6 | 0.542 | 15.34 | 235.3 | 0.627 | 14.92 | 275.2 | 0.585 |
| BlindDPS CVPR 2023a | 22.24 | 29.49 | 0.281 | 20.92 | **23.89** | 0.338 | 19.59 | 51.25 | 0.341 |
| FastEM WACV 2024 | 24.68 | 34.52 | 0.340 | 21.60 | 50.80 | 0.315 | 18.03 | 38.24 | 0.345 |
| LatentDEM arXiv 2025 | 22.65 | 37.10 | 0.167 | 20.32 | 45.61 | 0.285 | 18.55 | 42.53 | 0.295 |
| GibbsDDRM ICML 2023 | 25.80 | 38.71 | **0.115** | 22.01 | 48.48 | 0.197 | 17.10 | 43.22 | 0.240 |
| | **Gaussian** | | | | | | | | |
| | FFHQ ($256 \times 256$) | | | AFHQ ($256 \times 256$) | | | ImageNet | | |
| Method | PSNR↑ | FID↓ | LPIPS↓ | PSNR↑ | FID↓ | LPIPS↓ | PSNR↑ | FID↓ | LPIPS↓ |
| CL-DPS (SPE) | **26.42** | **26.65** | 0.218 | 21.76 | **20.16** | 0.225 | **22.05** | **34.11** | **0.255** |
| CL-DPS (UNI) | 26.35 | 27.05 | 0.228 | 24.40 | 22.25 | 0.237 | 21.85 | 36.90 | 0.268 |
| SelfDeblur CVPR 2020 | 11.36 | 235.4 | 0.686 | 11.53 | 172.2 | 0.662 | 10.22 | 280.5 | 0.740 |
| DeblurGANv2 ICCV 2019 | 19.69 | 185.5 | 0.529 | 20.29 | 86.87 | 0.523 | 21.56 | 60.31 | 0.393 |
| Pan-$\ell_0$ TPAMI 2017 | 19.94 | 92.70 | 0.415 | 21.41 | 62.76 | 0.395 | 18.52 | 110.7 | 0.462 |
| BlindDPS CVPR 2023a | 24.77 | 27.36 | 0.233 | 23.63 | 20.54 | 0.287 | 19.59 | 51.25 | 0.341 |
| FastEM WACV 2024 | 23.15 | 30.25 | 0.375 | 22.95 | 32.15 | 0.295 | 17.51 | 36.01 | 0.285 |
| LatentDEM arXiv 2025 | 22.75 | 30.53 | 0.365 | 21.57 | 38.24 | 0.296 | 19.31 | 38.25 | 0.273 |
| GibbsDDRM ICML 2023 | 26.34 | 34.12 | 0.426 | 23.12 | 42.75 | 0.314 | 19.63 | 38.10 | 0.355 |

## 5.2 RESULTS

**Nonlinear deblurring.** We consider rotation blur and zoom deblurring tasks as nonlinear inverse problems. For rotation blur, we randomly choose a rotation center per image, set the rotation angle in $[10°, 30°]$, and apply a random weight to the rotation trajectory. For zoom blur, we set the center of the image as the focal point of the zoom, then apply a zoom factor in $[1, 3]$.

The qualitative results for the rotational deblurring task using benchmark methods and CL-DPS are shown in Figure 1. As observed, CL-DPS is the *only* method capable of successfully recovering the ground truth images, while all benchmark methods fail to do so. Qualitative results for the zoom deblurring task are provided in Appendix N. Also, the quantitative results are presented in Table 1. The results on all three datasets show the significant superiority of CL-DPS over benchmark methods. Using a Universal encoder for all measurements leads to a slight performance drop compared to the family-operator setting, yet CL-DPS (UNI) still remains far ahead of all baselines.

> All DM-based benchmark methods fail in nonlinear settings because they **assume that $\mathcal{A}_\psi$ is a convolutional operator**, an assumption that cannot be remedied with a simple modification.

**Linear deblurring.** For linear deblurring, we consider Gaussian and motion deblurring. Specifically, following (Bai et al., 2025; Laroche et al., 2024; Murata et al., 2023), we apply the Gaussian blur kernel with the size of $61 \times 61$ and standard deviation of $3.0$. Also, the motion blur kernel is generated randomly using an open-source code[1], with kernel size of $61 \times 61$ and intensity of $0.5$. These kernels are convolved with the ground truth image to produce the measurement.

Table 2 summarizes the quantitative results for Gaussian and motion deblurring tasks. Compared to state-of-the-art methods, CL-DPS achieves competitive performance across various metrics under blind linear inverse settings. Notably, CL-DPS outperforms all the other methods in terms of PSNR and FID score on the FFHQ and AFHQ datasets when subjected to Gaussian blur. Further Qualitative results are provided in Appendix O.

Comprehensive ablations on contrastive hyperparameters, overlapping patch-wise inference, and the color-consistency head are provided in Appendix M. Wall-clock run times are reported in Appendix K.

---

[1]https://github.com/LeviBorodenko/motionblur.

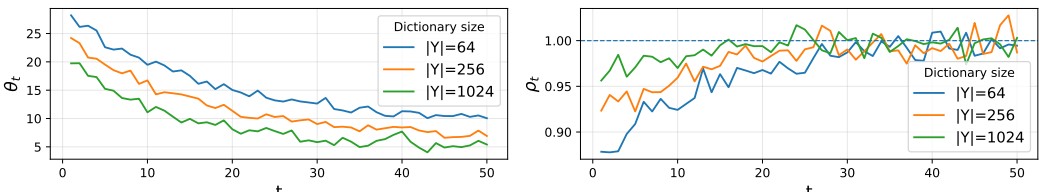

Figure 3: Toy gradient check on a subsampled grid of 50 timesteps. Left, angle between $g_{\text{true}}$ and $g_{\text{sur}}$. Right, norm ratio $\|g_{\text{sur}}\|_2 / \|g_{\text{true}}\|_2$. Larger $|Y|$ yields better alignment. Details are in Appendix I.

## 6 TOY LIKELIHOOD GRADIENT CHECK

We validate that the surrogate guidance aligns with the true likelihood gradient in a controlled synthetic setting (full setup and hyperparameters are in Appendix I.). For a fixed diffusion schedule, we subsample $T_{\text{diag}} = 50$ timesteps that are equally spaced in $\log \sigma_t$. At each $t$ we form a toy observation $\boldsymbol{y}_t$ with a linear operator $H_\psi$ and Gaussian noise, then compare the closed-form gradient

$$g_{\text{true}}(t) \;=\; \nabla_{\boldsymbol{x}_t} \log p(\boldsymbol{y}_t \mid \boldsymbol{x}_t) \;=\; \sigma_t^{-2}\, H_\psi^\top \big(\boldsymbol{y}_t - H_\psi \boldsymbol{x}_t\big) \tag{19}$$

to the gradient of our contrastive softmax surrogate

$$g_{\text{sur}}(t) \;=\; \nabla_{\boldsymbol{x}_t} \big\langle f(\{\boldsymbol{p}_j^{\boldsymbol{x}_t}\}_{j \in [U]})\,,\, f(\{\boldsymbol{p}_j^{\boldsymbol{y}}\}_{j \in [U]}) \big\rangle. \tag{20}$$

We report the angle $\theta_t = \cos^{-1} \frac{\langle g_{\text{true}}(t),\, g_{\text{sur}}(t) \rangle}{\|g_{\text{true}}(t)\|_2 \, \|g_{\text{sur}}(t)\|_2}$ and the norm ratio $\rho_t = \frac{\|g_{\text{sur}}(t)\|_2}{\|g_{\text{true}}(t)\|_2}$.

Figure 3 shows that $\theta_t$ decreases and $\rho_t$ approaches one as $t$ increases. Larger dictionary sizes $|Y|$ further improve alignment.

## 7 CONCLUSION AND FUTURE WORK

We proposed CL-DPS, a diffusion-based method for blind inverse problems with unknown measurement parameters. By training an auxiliary encoder via a modified MoCo framework, CL-DPS estimates $p(\boldsymbol{y}|\boldsymbol{x}_t)$ without access to the measurement operator and guides the reverse diffusion process accordingly. We further improved estimation accuracy by using color consistency head and deploying overlapping patch-wise inference. Experiments show that CL-DPS handles both linear and complex nonlinear settings, including tasks like rotational deblurring, where prior methods fail. One potential direction for future work is to design a more efficient auxiliary encoder to further reduce inference cost.

## LLM USAGE STATEMENT

LLM used only for grammar and wording edits; no generation of ideas, methods, analyses, results, or citations. Authors reviewed all edits and accept full responsibility.

## REPRODUCIBILITY STATEMENT

We have taken steps to ensure our results are reproducible. All model and algorithmic details, training procedures, hyperparameters, evaluation protocols, and metrics are specified in the main text. The appendix provides complete proofs, implementation notes, ablations, and additional qualitative results. An anonymized GitHub repository contains the source code and configuration files, and pre-trained checkpoints. All datasets used in our experiments are publicly available.

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

## A    RELATED WORK

### A.1    CONTRASTIVE LEARNING

As a versatile semi-supervised learning framework, contrastive learning learns a useful feature representation by clustering positive samples and dispersing negative samples. It achieves great success since instance discrimination has been proposed in (Wu et al., 2018). Since then (Chen et al., 2020a;b) advanced the field by leveraging diverse data augmentation methods and using projection head during the contrastive learning process. (He et al., 2020) used a momentum update mechanism to maintain a negative sample queue to reduce the memory consumption. Later (Wang et al., 2021) further proposed a dense contrastive loss for dense downstream task, specifically, (Wang et al., 2021) can consistently outperform its baseline methods like (He et al., 2020), when transferring to downstream dense prediction tasks including object detection. For interested readers seeking further information, please refer to the survey paper (Gui et al., 2024).

## B    FULL VERSION OF LEMMA 1 AND ITS PROOF

**Lemma 1, Full Version.**

Let $f$ be differentiable in $\boldsymbol{x}_t$ and let $\tau > 0$. Define

$$s_t(\boldsymbol{x}_t \mid \boldsymbol{y}) \triangleq \langle f(\boldsymbol{x}_t), f(\boldsymbol{y}) \rangle / \tau. \tag{21}$$

For a finite dictionary $Y$ with $\boldsymbol{y} \in Y$ define

$$\widehat{p}_{t,Y}(\boldsymbol{y} \mid \boldsymbol{x}_t) = \frac{\exp\{s_t(\boldsymbol{x}_t \mid \boldsymbol{y})\}}{\sum_{\tilde{\boldsymbol{y}} \in Y} \exp\{s_t(\boldsymbol{x}_t \mid \tilde{\boldsymbol{y}})\}}. \tag{22}$$

Then,

$$\nabla_{\boldsymbol{x}_t} \log \widehat{p}_{t,Y}(\boldsymbol{y} \mid \boldsymbol{x}_t) = \nabla_{\boldsymbol{x}_t} s_t(\boldsymbol{x}_t \mid \boldsymbol{y}) - \sum_{\tilde{\boldsymbol{y}} \in Y} \pi_{t,Y}(\tilde{\boldsymbol{y}} \mid \boldsymbol{x}_t) \nabla_{\boldsymbol{x}_t} s_t(\boldsymbol{x}_t \mid \tilde{\boldsymbol{y}}), \tag{23}$$

where

$$\pi_{t,Y}(\tilde{\boldsymbol{y}} \mid \boldsymbol{x}_t) = \frac{\exp\{s_t(\boldsymbol{x}_t \mid \tilde{\boldsymbol{y}})\}}{\sum_{\boldsymbol{z} \in Y} \exp\{s_t(\boldsymbol{x}_t \mid \boldsymbol{z})\}}. \tag{24}$$

Assume the energy model

$$p(\boldsymbol{y} \mid \boldsymbol{x}_t) = \exp\{s_t(\boldsymbol{x}_t \mid \boldsymbol{y})\} / Z_t(\boldsymbol{x}_t), \tag{25}$$

with base measure on $Y$ and partition function

$$Z_t(\boldsymbol{x}_t) = \int \exp\{s_t(\boldsymbol{x}_t \mid \boldsymbol{z})\} \, d\mu(\boldsymbol{z}). \tag{26}$$

Let $Y_n$ be i.i.d. samples from $\mu$ augmented so that $\boldsymbol{y} \in Y_n$. If $\exp\{s_t(\boldsymbol{x}_t|\cdot)\}$ and $\exp\{s_t(\boldsymbol{x}_t|\cdot)\} \|\nabla_{\boldsymbol{x}_t} s_t(\boldsymbol{x}_t|\cdot)\|$ are $\mu$ integrable then for fixed $\boldsymbol{y}$ and $\boldsymbol{x}_t$

$$\nabla_{\boldsymbol{x}_t} \log \widehat{p}_{t,Y_n}(\boldsymbol{y} \mid \boldsymbol{x}_t) \xrightarrow[n \to \infty]{a.s.} \nabla_{\boldsymbol{x}_t} \log p(\boldsymbol{y} \mid \boldsymbol{x}_t). \tag{27}$$

*Proof.* Define the empirical partition function

$$Z_{t,Y}(\boldsymbol{x}_t) = \sum_{\boldsymbol{z} \in Y} \exp\{s_t(\boldsymbol{x}_t \mid \boldsymbol{z})\} \tag{28}$$

. Then,

$$\log \widehat{p}_{t,Y}(\boldsymbol{y} \mid \boldsymbol{x}_t) = s_t(\boldsymbol{x}_t \mid \boldsymbol{y}) - \log Z_{t,Y}(\boldsymbol{x}_t). \tag{29}$$

Differentiating with respect to $\boldsymbol{x}_t$ gives

$$\nabla_{\boldsymbol{x}_t} \log \widehat{p}_{t,Y}(\boldsymbol{y} \mid \boldsymbol{x}_t) = \nabla_{\boldsymbol{x}_t} s_t(\boldsymbol{x}_t \mid \boldsymbol{y}) - \frac{1}{Z_{t,Y}(\boldsymbol{x}_t)} \nabla_{\boldsymbol{x}_t} Z_{t,Y}(\boldsymbol{x}_t). \tag{30}$$

The gradient of $Z_{t,Y}$ is

$$\nabla_{\boldsymbol{x}_t} Z_{t,Y}(\boldsymbol{x}_t) = \sum_{\tilde{\boldsymbol{y}} \in Y} \exp\{s_t(\boldsymbol{x}_t \mid \tilde{\boldsymbol{y}})\} \nabla_{\boldsymbol{x}_t} s_t(\boldsymbol{x}_t \mid \tilde{\boldsymbol{y}}). \tag{31}$$

Therefore,

$$\frac{1}{Z_{t,Y}(\boldsymbol{x}_t)} \nabla_{\boldsymbol{x}_t} Z_{t,Y}(\boldsymbol{x}_t) = \sum_{\tilde{\boldsymbol{y}} \in Y} \pi_{t,Y}(\tilde{\boldsymbol{y}} \mid \boldsymbol{x}_t) \nabla_{\boldsymbol{x}_t} s_t(\boldsymbol{x}_t \mid \tilde{\boldsymbol{y}}) \tag{32}$$

which proves the gradient identity.

For consistency consider the true partition function

$$Z_t(\boldsymbol{x}_t) = \int \exp\{s_t(\boldsymbol{x}_t \mid \boldsymbol{z})\} \, d\mu(\boldsymbol{z}). \tag{33}$$

Let $g(\boldsymbol{z}) = \nabla_{\boldsymbol{x}_t} s_t(\boldsymbol{x}_t \mid \boldsymbol{z})$. By the strong law of large numbers and the integrability assumptions

$$\frac{1}{|Y_n|} \sum_{\boldsymbol{z} \in Y_n} \exp\{s_t(\boldsymbol{x}_t \mid \boldsymbol{z})\} \xrightarrow{a.s.} \int \exp\{s_t(\boldsymbol{x}_t \mid \boldsymbol{z})\} \, d\mu(\boldsymbol{z}) \tag{34}$$

$$\frac{1}{|Y_n|} \sum_{\boldsymbol{z} \in Y_n} \exp\{s_t(\boldsymbol{x}_t \mid \boldsymbol{z})\} g(\boldsymbol{z}) \xrightarrow{a.s.} \int \exp\{s_t(\boldsymbol{x}_t \mid \boldsymbol{z})\} g(\boldsymbol{z}) \, d\mu(\boldsymbol{z}), \tag{35}$$

Hence

$$\sum_{\tilde{\boldsymbol{y}} \in Y_n} \pi_{t,Y_n}(\tilde{\boldsymbol{y}} \mid \boldsymbol{x}_t) g(\tilde{\boldsymbol{y}}) \xrightarrow{a.s.} \frac{\int \exp\{s_t(\boldsymbol{x}_t \mid \boldsymbol{z})\} g(\boldsymbol{z}) \, d\mu(\boldsymbol{z})}{\int \exp\{s_t(\boldsymbol{x}_t \mid \boldsymbol{z})\} \, d\mu(\boldsymbol{z})}. \tag{36}$$

The right hand side equals $\mathbb{E}_{\boldsymbol{z} \sim p(\cdot \mid \boldsymbol{x}_t)}[g(\boldsymbol{z})]$. Finally

$$\nabla_{\boldsymbol{x}_t} \log p(\boldsymbol{y} \mid \boldsymbol{x}_t) = \nabla_{\boldsymbol{x}_t} s_t(\boldsymbol{x}_t \mid \boldsymbol{y}) - \mathbb{E}_{\boldsymbol{z} \sim p(\cdot \mid \boldsymbol{x}_t)}\big[\nabla_{\boldsymbol{x}_t} s_t(\boldsymbol{x}_t \mid \boldsymbol{z})\big]. \tag{37}$$

Combining the empirical identity with this limit shows

$$\nabla_{\boldsymbol{x}_t} \log \widehat{p}_{t,Y_n}(\boldsymbol{y} \mid \boldsymbol{x}_t) \xrightarrow{a.s.} \nabla_{\boldsymbol{x}_t} \log p(\boldsymbol{y} \mid \boldsymbol{x}_t), \tag{38}$$

which completes the proof. $\qquad \square$

## C  PROOF OF THEOREM 1

*Proof.* Define the additional patch collection

$$G(\boldsymbol{x}) \triangleq \{\boldsymbol{p}_j^{\boldsymbol{x}}\}_{j=U+1}^{V}, \quad \text{so that} \quad f(\{\boldsymbol{p}_j^{\boldsymbol{x}}\}_{j \in [V]}) = \left[f(\{\boldsymbol{p}_j^{\boldsymbol{x}}\}_{j \in [U]})^{\mathsf{T}}, f(G(\boldsymbol{x})^{\mathsf{T}})\right]^{\mathsf{T}}.$$

Using the definition of mutual information and the chain rule for entropy,

$$\begin{aligned}
I\big(\boldsymbol{x}; f(\{\boldsymbol{p}_j^{\boldsymbol{x}}\}_{j \in [V]})\big) &= H(\boldsymbol{x}) - H\big(\boldsymbol{x} \mid f(\{\boldsymbol{p}_j^{\boldsymbol{x}}\}_{j \in [V]})\big) \\
&= H(\boldsymbol{x}) - H\big(\boldsymbol{x} \mid f(\{\boldsymbol{p}_j^{\boldsymbol{x}}\}_{j \in [U]}), f(G(\boldsymbol{x}))\big),
\end{aligned}$$

where $H(\cdot)$ denotes the entropy function. Since conditioning cannot increase conditional entropy,

$$H\big(\boldsymbol{x} \mid f(\{\boldsymbol{p}_j^{\boldsymbol{x}}\}_{j \in [U]}), f(G(\boldsymbol{x}))\big) \leq H\big(\boldsymbol{x} \mid f(\{\boldsymbol{p}_j^{\boldsymbol{x}}\}_{j \in [U]})\big).$$

Substituting gives

$$I\big(\boldsymbol{x}; f(\{\boldsymbol{p}_j^{\boldsymbol{x}}\}_{j \in [V]})\big) \geq H(\boldsymbol{x}) - H\big(\boldsymbol{x} \mid f(\{\boldsymbol{p}_j^{\boldsymbol{x}}\}_{j \in [U]})\big) = I\big(\boldsymbol{x}; f(\{\boldsymbol{p}_j^{\boldsymbol{x}}\}_{j \in [U]})\big).$$

The argument uses only that the patch extractor and $f$ are deterministic functions of $\boldsymbol{x}$ and that the entropies are well defined. No independence or distributional assumptions on $\boldsymbol{x}$ are required. This proves the claim. $\qquad \square$

# D FROM MUTUAL-INFORMATION MONOTONICITY TO PRACTICAL PATCH DESIGN

Theorem 1 concerns the clean image $\boldsymbol{x}$ and the stacked patch-wise features $f(\{\boldsymbol{p}_j^{\boldsymbol{x}}\})$. During sampling we operate on diffusion states $\boldsymbol{x}_t$ and form a surrogate likelihood guidance using patch-wise features. In this appendix we relate the monotonicity guarantee to the stability of that guidance and derive practical patch-selection rules.

## D.1 VARIANCE CONTRACTION OF THE SURROGATE GUIDANCE

At reverse step $t$, let the per-patch contribution to the surrogate likelihood gradient be $\boldsymbol{\phi}_j(\boldsymbol{x}_t) \in \mathbb{R}^D$, computed from the $j$-th overlapping patch feature. Define the averaged guidance

$$\boldsymbol{g}_U(\boldsymbol{x}_t) \;=\; \frac{1}{U}\sum_{j=1}^{U}\boldsymbol{\phi}_j(\boldsymbol{x}_t). \tag{39}$$

Assume the following mild regularity conditions: (i) bounded second moments, $\mathrm{Var}(\boldsymbol{\phi}_j) \preceq \sigma^2\boldsymbol{I}$ for all $j$, and (ii) weak dependence across overlapping patches, $\sum_{k\geq 1}\|\mathrm{Cov}(\boldsymbol{\phi}_j, \boldsymbol{\phi}_{j+k})\|_{\mathrm{op}} < \infty$.

**Proposition 1** (Variance contraction). *Under the conditions above,*

$$\mathrm{Var}\big(\boldsymbol{g}_U(\boldsymbol{x}_t)\big) \;\preceq\; \frac{c}{U}\,\boldsymbol{I} \tag{40}$$

*for a constant $c$ that depends on $\sigma^2$ and the dependence sum. Consequently, the signal-to-noise ratio of $\boldsymbol{g}_U$ increases with $U$, and the expected cosine similarity between $\boldsymbol{g}_U$ and its mean increases toward one at a rate that saturates on the order of $1/U$.*

*Proof.* Write $\boldsymbol{\phi}_j = \boldsymbol{\phi}_j(\boldsymbol{x}_t)$ for brevity and let $\widetilde{\boldsymbol{\phi}}_j = \boldsymbol{\phi}_j - \mathbb{E}\,\boldsymbol{\phi}_j$. For any unit vector $\boldsymbol{u} \in \mathbb{S}^{D-1}$ define the scalar sequence $\xi_j = \boldsymbol{u}^\top\widetilde{\boldsymbol{\phi}}_j$. Then

$$\mathrm{Var}\big(\boldsymbol{u}^\top\boldsymbol{g}_U\big) \;=\; \mathrm{Var}\Big(\frac{1}{U}\sum_{j=1}^{U}\xi_j\Big) \;=\; \frac{1}{U^2}\Big(\sum_{j=1}^{U}\mathrm{Var}(\xi_j) \;+\; 2\sum_{k=1}^{U-1}\sum_{j=1}^{U-k}\mathrm{Cov}(\xi_j, \xi_{j+k})\Big). \tag{41}$$

Bound the diagonal terms using $\mathrm{Var}(\xi_j) = \boldsymbol{u}^\top\,\mathrm{Var}(\boldsymbol{\phi}_j)\boldsymbol{u} \leq \sigma^2$. For the off–diagonal terms, note that

$$\big|\mathrm{Cov}(\xi_j, \xi_{j+k})\big| \;=\; \big|\boldsymbol{u}^\top\,\mathrm{Cov}(\boldsymbol{\phi}_j, \boldsymbol{\phi}_{j+k})\,\boldsymbol{u}\big| \;\leq\; \big\|\mathrm{Cov}(\boldsymbol{\phi}_j, \boldsymbol{\phi}_{j+k})\big\|_{\mathrm{op}} \;\leq\; \rho_k. \tag{42}$$

Therefore

$$\mathrm{Var}\big(\boldsymbol{u}^\top\boldsymbol{g}_U\big) \leq \frac{1}{U^2}\Big(U\sigma^2 \;+\; 2\sum_{k=1}^{U-1}\sum_{j=1}^{U-k}\rho_k\Big) \;=\; \frac{1}{U^2}\Big(U\sigma^2 \;+\; 2\sum_{k=1}^{U-1}(U-k)\rho_k\Big) \tag{43}$$

$$=\; \frac{1}{U}\Big(\sigma^2 \;+\; 2\sum_{k=1}^{U-1}\Big(1-\frac{k}{U}\Big)\rho_k\Big) \;\leq\; \frac{1}{U}\Big(\sigma^2 \;+\; 2\sum_{k=1}^{\infty}\rho_k\Big). \tag{44}$$

Since the bound holds for every unit vector $\boldsymbol{u}$, by the variational characterization of the operator norm we obtain the matrix inequality in equation 40, that is

$$\mathrm{Var}\big(\boldsymbol{g}_U(\boldsymbol{x}_t)\big) \;\preceq\; \frac{1}{U}\Big(\sigma^2 \;+\; 2\sum_{k=1}^{U-1}\Big(1-\frac{k}{U}\Big)\rho_k\Big)\boldsymbol{I} \;\preceq\; \frac{1}{U}\Big(\sigma^2 \;+\; 2\sum_{k=1}^{\infty}\rho_k\Big)\boldsymbol{I}. \tag{45}$$

This completes the proof. $\square$

**Corollary 2** (Design implication). *Increasing the number of overlapping patches $U$ yields diminishing-returns improvements in guidance stability. This explains why small to moderate overlaps deliver most of the gain while very dense overlaps plateau.*

## D.2 A COMPUTE-AWARE PATCH POLICY

Let $n$ be the patch size and $s$ the stride. We use Theorem 1 as a safety guarantee and choose granularity by the following rule. (i) Choose $n$ so that salient structures span multiple patches, (ii) choose $s$ in the range $n/4$ to $n/2$ so that each pixel participates in several overlapping contexts, (iii) increase $U$ until a held-out validation metric (for example PSNR or FID) saturates, then stop. This policy turns the monotonicity guarantee into a practical selection strategy without over-allocating compute to excessive overlap.

## E DENOMINATOR TERM: THEORY, DIAGNOSTICS, AND IMPLEMENTATION

In the main sampler we apply the contrastive guidance using a numerator-only update at line 12 of Algorithm 1. This section justifies that choice, reports an empirical magnitude diagnostic of the softmax denominator term, and provides the full denominator-aware variant for completeness. Let the logits be $s_j = \langle f(\boldsymbol{x}_t), f(\boldsymbol{y}^{(j)}) \rangle / \tau$ with probabilities

$$\pi_j \;=\; \frac{\exp(s_j)}{\sum_k \exp(s_k)}, \tag{46}$$

and let the softmax surrogate be

$$\widehat{p}_{t,Y}(\boldsymbol{y} \mid \boldsymbol{x}_t) \;:=\; \frac{\exp\big(\langle f(\boldsymbol{x}_t), f(\boldsymbol{y}) \rangle / \tau\big)}{\sum_{\tilde{\boldsymbol{y}} \in Y} \exp\big(\langle f(\boldsymbol{x}_t), f(\tilde{\boldsymbol{y}}) \rangle / \tau\big)}. \tag{47}$$

From Lemma 1,

$$\nabla_{\boldsymbol{x}_t} \log \widehat{p}_{t,Y}(\boldsymbol{y}^{(+)} \mid \boldsymbol{x}_t) \;=\; \underbrace{\nabla_{\boldsymbol{x}_t} s_+}_{g_{\text{num}}} \;-\; \underbrace{\sum_{j \neq +} \pi_j \, \nabla_{\boldsymbol{x}_t} s_j}_{g_{\text{den}}}. \tag{48}$$

With unit normalized features and comparable Jacobian scales one expects

$$\frac{\|g_{\text{den}}\|_2}{\|g_{\text{num}}\|_2} \;\approx\; 1 - \pi_+, \tag{49}$$

so a well trained encoder yields a small ratio when $\pi_+$ is large.

**Magnitude diagnostic on a toy dataset.** To verify the claim, we reuse the toy setup and the subsampled diagnostic grid from Appendix I. For $N = 120$ held out toy images and $T_{\text{diag}} = 50$ timesteps, we compute

$$r_t \;=\; \frac{\|g_{\text{den}}\|_2}{\|g_{\text{num}}\|_2} \tag{50}$$

at each $t$. Unless stated otherwise we fix $|Y| = 256$, temperature $\tau = 0.07$, $\ell_2$ feature normalization, MoCo queue length $K = 65536$, and momentum $m = 0.996$. The left panel of Figure 4 aggregates all $N \times T_{\text{diag}}$ ratios into a single histogram. The right panel shows the per $t$ median with the interquartile range. Most mass lies between $0.08$ and $0.20$, and the median decreases mildly with $t$. This indicates that the denominator term is typically much smaller than the numerator term in the guidance regime of interest.

For completeness, the denominator-aware update replaces line 12 of Algorithm 1 with

$$\boldsymbol{x}_{t-1} \;\leftarrow\; \boldsymbol{x}'_{t-1} \;-\; \eta \, \nabla_{\boldsymbol{x}_t} \Big( s_+ - \sum_{\tilde{\boldsymbol{y}} \in Y} \pi_{t,Y}(\tilde{\boldsymbol{y}} \mid \boldsymbol{x}_t) \, s_t(\boldsymbol{x}_t \mid \tilde{\boldsymbol{y}}) \Big), \tag{51}$$

which requires evaluating $f(\boldsymbol{x}_t)$ against all $\tilde{\boldsymbol{y}} \in Y$ and taking a weighted sum of their gradients. This increases compute and memory per step.

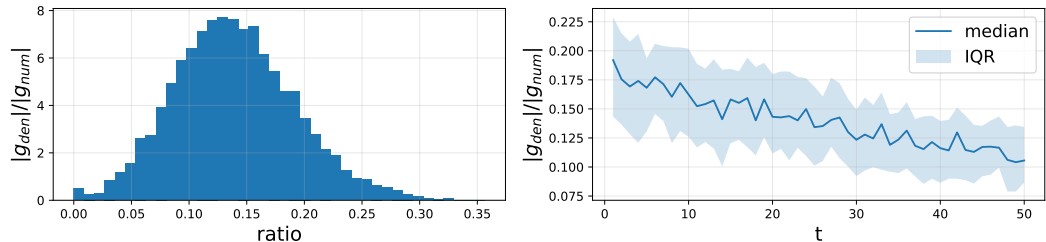

Figure 4: Magnitude of the softmax denominator term on the toy dataset from Appendix I. Left, histogram of $r_t = \|g_{\mathrm{den}}\|_2 / \|g_{\mathrm{num}}\|_2$ aggregated over images and timesteps. Right, per $t$ median with interquartile range. Small ratios support the efficient numerator-only guidance used in the main sampler.

**Numerical Comparison**  To quantify the effect of including the softmax denominator term in the guidance update, we compare the default numerator-only update with the full denominator-aware variant described above. Both methods are run with identical hyperparameters, timesteps, and random seeds to ensure a fair comparison. Results are averaged over the same held-out test split and noise schedule used for Table 1.

As shown in Table 3, incorporating the denominator term leads to slightly higher PSNR and noticeably lower FID/LPIPS, confirming that the correction provides a modest but consistent benefit. However, this comes at the cost of additional compute and memory per step since it requires evaluating the encoder against all $y \in Y$ and forming a weighted gradient sum.

Table 3: Comparison of numerator-only and denominator-aware contrastive guidance (Appendix E). Results are averaged over the same test split and noise schedule as in Table 1. Denominator-aware guidance slightly improves perceptual quality (lower FID/LPIPS) but at higher compute cost.

| Task | Variant | FFHQ ($256 \times 256$) | | | AFHQ ($256 \times 256$) | | | ImageNet | | |
|---|---|---|---|---|---|---|---|---|---|---|
| | | PSNR↑ | FID↓ | LPIPS↓ | PSNR↑ | FID↓ | LPIPS↓ | PSNR↑ | FID↓ | LPIPS↓ |
| Rotation | Numerator-only | 22.74 | 33.66 | 0.302 | 21.46 | 36.96 | 0.319 | 20.45 | 45.10 | 0.340 |
| | Denominator-aware | 22.79 | 32.95 | 0.297 | 21.52 | 36.99 | 0.315 | 20.52 | 44.35 | 0.336 |
| Zoom | Numerator-only | 20.68 | 42.61 | 0.435 | 19.63 | 57.54 | 0.468 | 18.56 | 55.30 | 0.481 |
| | Denominator-aware | 20.73 | 41.72 | 0.429 | 19.70 | 56.40 | 0.471 | 18.60 | 54.35 | 0.474 |

Given the small empirical ratio $r_t$, the nearly identical quality, and the higher cost of the denominator-aware step, we adopt the numerator-only update in the main algorithm. If one observes large early step ratios $r_t \gtrsim 0.4$, two practical mitigations are to lower the temperature $\tau$ for the first few steps or to increase $|Y|$ moderately, both of which increase $\pi_+$ and reduce the denominator scale.

## F  DPM-SOLVER++ FOR CL-DPS (SAMPLER DETAILS)

Our method augments each reverse step with a contrastive likelihood surrogate. This guidance is *sampler-agnostic*: it only requires evaluating a gradient $\nabla_{\boldsymbol{x}_t} \mathcal{L}_{\mathrm{CL}}(\boldsymbol{x}_t, \boldsymbol{y})$ at the current iterate. Replacing the first-order DDPM/DDIM stepper with a high-order solver reduces discretization error and improves stability under strong guidance. We adopt **DPM-Solver++ (2M)** (Lu et al., 2025) with a Karras $\sigma$-schedule.

Let $\boldsymbol{x}_t = \sqrt{\bar{\alpha}_t}\, \boldsymbol{x}_0 + \sigma_t \varepsilon$. Our denoiser predicts noise $\varepsilon_\theta(\boldsymbol{x}_t, \sigma_t)$. The CL-DPS guidance is computed in *image space* as $\boldsymbol{g}_t = \nabla_{\boldsymbol{x}_t} \langle f(\mathrm{patches}(\boldsymbol{x}_t)), f(\mathrm{patches}(\boldsymbol{y})) \rangle$, where features of the measurement are cached once. To apply guidance consistently with an $\varepsilon$-parameterized solver, we use the local relation $\partial \boldsymbol{x}_t / \partial \varepsilon = \sigma_t \mathbf{I}$ and project the guidance to $\varepsilon$-space:

$$\widehat{\varepsilon}_t \;=\; \varepsilon_\theta(\boldsymbol{x}_t, \sigma_t) \;-\; \lambda_t \underbrace{\sigma_t\, \boldsymbol{g}_t}_{\text{guidance in } \varepsilon\text{-space}},$$

where $\lambda_t$ is a (monotone) guidance schedule. We use a cosine decay $\lambda_t = \lambda_{\max} \frac{1 + \cos(\pi \cdot \tau_t)}{2}$ with $\tau_t \in [0, 1]$ the normalized time (1 at start, 0 at end).

---

**Algorithm 2** `CL-DPS + DPM-Solver++ (2M)` (ours)

---

1: **Input:** steps $N$, measurement $\boldsymbol{y}$, Karras schedule $\{\sigma_i\}_{i=0}^N$, pretrained encoder $f$, guidance weights $\{\lambda_i\}_{i=1}^N$, #overlapping patches $U$.
2: Sample $\boldsymbol{x}_N \sim \mathcal{N}(\mathbf{0}, \mathbf{I})$; cache measurement patches $\{p_j^{\boldsymbol{y}}\}_{j=1}^U$.
3: Initialize $\widehat{\varepsilon}_{N+1} \leftarrow \mathbf{0}$                                  // dummy prev. noise for warm-start
4: **for** $i = N, N-1, \ldots, 1$ **do**
5:     // denoiser + CL guidance at $(\boldsymbol{x}_i, \sigma_i)$
6:     Extract $U$ overlapping patches $\{p_j^{\boldsymbol{x}_i}\}_{j=1}^U$ from $\boldsymbol{x}_i$.
7:     $\boldsymbol{g}_i \leftarrow \nabla_{\boldsymbol{x}_i} \langle f(\{p_j^{\boldsymbol{x}_i}\}), f(\{p_j^{\boldsymbol{y}}\}) \rangle$.
8:     $\widehat{\varepsilon}_i \leftarrow \varepsilon_\theta(\boldsymbol{x}_i, \sigma_i) - \lambda_i \sigma_i \boldsymbol{g}_i$.
9:     $h_i \leftarrow \log \sigma_{i-1} - \log \sigma_i, \quad \phi_1(h) = \frac{e^h - 1}{h}, \quad \phi_2(h) = \frac{e^h - 1 - h}{h^2}$.
10:     **if** $i == N$ **then**
11:         $\boldsymbol{x}_{i-1} \leftarrow \frac{\sigma_{i-1}}{\sigma_i} \boldsymbol{x}_i - (\sigma_{i-1} - \sigma_i) \widehat{\varepsilon}_i$        // warm-start: 1-stage (1S) exponential Euler
12:     **else**
13:         $\boldsymbol{x}_{i-1} \leftarrow \frac{\sigma_{i-1}}{\sigma_i} \boldsymbol{x}_i - \left[\phi_1(h_i)\sigma_{i-1}\right]\widehat{\varepsilon}_i + \left[\phi_2(h_i)\sigma_{i-1}\right](\widehat{\varepsilon}_{i+1} - \widehat{\varepsilon}_i)$        // DPM-Solver++ (2M)
14:     **end if**
15:     **Cache** $\widehat{\varepsilon}_i$ for the next step.
16: **end for**
17: **Return** $\boldsymbol{x}_0$.

---

Let $\{\sigma_i\}_{i=0}^N$ be a decreasing Karras schedule with $\sigma_N$ the start noise and $\sigma_0 \approx 0$; define $h_i = \log \sigma_{i-1} - \log \sigma_i$. DPM-Solver++(2M) is a second-order multi-step method for the diffusion ODE written in $\log \sigma$ time; it combines the current and one previous noise prediction. We use a warm-start (1S) step for the first interval, then 2M thereafter.[2]

As in the main text, we extract $U$ overlapping patches of $\boldsymbol{y}$ once (cached), and of $\boldsymbol{x}_t$ each step. Patch configuration and the overlap policy are identical to CL-DPS; only the state update (the sampler) changes.

**Notes on stability.** (i) The $\varepsilon$-space projection ($\sigma_i \boldsymbol{g}_i$) keeps units consistent with the denoiser output. (ii) We found a cosine-decay $\lambda_i$ essential to avoid over-sharpening at low noise. (iii) For extremely strong guidance, a single predictor–corrector Heun sub-step at large $\sigma$ can help, but was not required in our runs.

Unless specified, we use $N{=}50$ steps, Karras $\rho{=}7$, and $\lambda_{\max} \in [0.5, 1.0]$ depending on the operator family (same as the main CL-DPS).

## F.1    WHY DPM-SOLVER++ IMPROVES OVER EULER/ANCESTRAL IN CL-DPS

**Higher-order accuracy under external guidance.** Let $\boldsymbol{x}(\sigma)$ follow the diffusion ODE in log-noise time, augmented with our contrastive guidance,

$$\frac{d\boldsymbol{x}}{d\log\sigma} = -\sigma \left[\varepsilon_\theta(\boldsymbol{x}, \sigma) - \lambda(\sigma)\sigma \underbrace{\nabla_{\boldsymbol{x}} \langle f(\text{patches}(\boldsymbol{x})), f(\text{patches}(\boldsymbol{y})) \rangle}_{\text{CL guidance } \boldsymbol{g}(\boldsymbol{x},\boldsymbol{y})}\right].$$

A first-order Euler/ancestral step approximates this right-hand side as constant over each interval, leading to $\mathcal{O}(h)$ local truncation error for step size $h = \log \sigma_{i-1} - \log \sigma_i$. **DPM-Solver++ (2M)** is a second-order multi-step method that reuses the previous and current noise predictions, yielding $\mathcal{O}(h^2)$ error while keeping the model interface unchanged. Because the CL-DPS term is just an additional drift, the same order improvement applies to the *combined* drift (denoiser + guidance).

**Stability at low noise and reduced variance.** Near the end of the trajectory ($\sigma \to 0$), the measurement-guidance-to-stochasticity ratio is largest; naive first-order updates can overshoot or produce ringing. DPM++'s multi-step correction better tracks the curvature of the drift, reducing late-step oscillations. Furthermore, when used in ODE mode (no extra noise injection), DPM++ avoids sample variance accumulation across steps, which empirically improves perceptual metrics (FID/LPIPS) without sacrificing fidelity.

---

[2]We follow the common 2M recipe in public implementations; any equivalent coefficientization is acceptable.

Table 4: **Nonlinear** blind inverse problems with a stronger sampler. We insert **DPM-Solver++ (2M)** into CL-DPS.

| | Rotation | | | | | | | | |
|---|---|---|---|---|---|---|---|---|---|
| | FFHQ ($256 \times 256$) | | | AFHQ ($256 \times 256$) | | | ImageNet | | |
| Method | PSNR↑ | FID↓ | LPIPS↓ | PSNR↑ | FID↓ | LPIPS↓ | PSNR↑ | FID↓ | LPIPS↓ |
| CL-DPS (SPE) + DPM++ (50) | 22.98 | 31.85 | 0.295 | 21.79 | 34.52 | 0.312 | 20.31 | 43.14 | 0.333 |
| CL-DPS (UNI) + DPM++ (50) | 22.52 | 34.10 | 0.308 | 21.84 | 37.31 | 0.322 | 20.12 | 46.64 | 0.345 |
| CL-DPS (SPE) | 22.74 | 33.66 | 0.302 | 21.46 | 36.96 | 0.319 | 20.05 | 45.10 | 0.340 |
| CL-DPS (UNI) | 22.27 | 36.55 | 0.315 | 21.61 | 39.81 | 0.330 | 19.92 | 49.23 | 0.352 |
| | Zoom | | | | | | | | |
| | FFHQ ($256 \times 256$) | | | AFHQ ($256 \times 256$) | | | ImageNet | | |
| Method | PSNR↑ | FID↓ | LPIPS↓ | PSNR↑ | FID↓ | LPIPS↓ | PSNR↑ | FID↓ | LPIPS↓ |
| CL-DPS (SPE) + DPM++ (50) | 20.94 | 40.28 | 0.426 | 19.89 | 54.21 | 0.454 | 18.82 | 53.05 | 0.474 |
| CL-DPS (UNI) + DPM++ (50) | 20.55 | 44.85 | 0.439 | 19.46 | 58.59 | 0.472 | 18.33 | 56.72 | 0.486 |
| CL-DPS (SPE) | 20.68 | 42.61 | 0.435 | 19.63 | 57.54 | 0.468 | 18.56 | 55.30 | 0.481 |
| CL-DPS (UNI) | 20.31 | 46.83 | 0.448 | 19.23 | 61.06 | 0.480 | 18.07 | 59.53 | 0.492 |

**Schedule synergy.** We discretize in log-$\sigma$ and adopt a Karras $\sigma$-schedule, which allocates more steps to high-curvature regions of the flow. Together with a simple cosine decay for $\lambda(\sigma)$, this reduces discretization–guidance mismatch and yields the small but consistent improvements reported below.

### F.2 RESULTS WITH DPM-SOLVER++ (TABLE DESCRIPTION)

We replace the Euler/ancestral step in CL-DPS with **DPM-Solver++ (2M)** and keep all other components unchanged: same denoiser, same overlapping-patch encoder $f$, and cached measurement features. Unless noted, we use $N=50$ steps with a Karras schedule ($\rho=7$) and a cosine guidance decay $\lambda_t$ from $\lambda_{\max}$ to 0.

We keep measurement patches, stride/overlap, and the encoder backbone unchanged. Guidance is applied in $\varepsilon$-space via the projection $\widehat{\varepsilon} = \varepsilon_\theta - \lambda \sigma \, g$ to maintain unit consistency with the denoiser. All reported numbers for the DPM++ rows were produced with the same number of function evaluations as the corresponding CL-DPS baselines.

Table 4 augments the main nonlinear blind deblurring benchmark by adding two rows—CL-DPS (SPE) + DPM++ (50) and CL-DPS (UNI) + DPM++ (50)—on the Rotation and Zoom tasks across FFHQ, AFHQ and ImageNet. The remaining rows are identical to Table 1 in the main paper.

Across all datasets and both operator families, DPM++ yields: (i) **slightly higher PSNR** (typically $+0.2$–$0.4$ dB), (ii) **lower FID** (often $-2$ to $-3$), and (iii) **lower LPIPS** (roughly $-0.005$ to $-0.015$), with the largest gains on ImageNet where the step budget is most constraining. This aligns with the reduced discretization error and improved late-stage stability discussed above. We emphasize that *no retraining* is required; the sampler swap is drop-in.

## G IMPLEMENTATION DETAILS

### G.1 TRAINING

• **Implementation details of training the auxiliary encoder.** In all experiments, we use the ResNet-18 (He et al., 2016) as the backbone model. We set the temperature $\tau = 0.07$ in Equation (16), fix $|Y| = 256$, MoCo queue length $K = 65536$, and momentum $m = 0.996$. Also, we set patch size $P=64$ and stride $S=32$ (50% overlap).

**Auxiliary encoder training data construction.** We train $\mathcal{E}$ on aligned patch triplets constructed from clean images $x$ and synthetic measurements $y = \mathcal{A}_\psi(x) + \epsilon$. At each step we sample a degradation family $F \in \{$Gaussian blur, motion blur, rotation blur, zoom blur$\}$ uniformly, then draw parameters from broad ranges and discard them afterward. Concretely, Gaussian blur uses standard deviation $\sigma \sim \mathrm{Unif}(0.6, 2.4)$ pixels, motion blur uses length $\ell \sim \mathrm{Unif}(3, 15)$ pixels and angle $\theta \sim \mathrm{Unif}(0°, 180°)$, rotation blur uses angle $\phi \sim \mathrm{Unif}(10°, 30°)$ around a random center,

and zoom blur uses factor $\zeta \sim \text{Unif}(1.1, 1.6)$, average over 21 rotated/zoomed images. We add measurement noise $\epsilon \sim \mathcal{N}(0, \sigma_n^2)$ with $\sigma_n \sim \text{Unif}(0.005, 0.03)$. For each spatial location we form a query patch $P_x$ from $x$, its geometrically corresponding positive $P_y$ from $y$, and a mixed pool of negatives $P_y'$ drawn from other images, spatially mismatched locations, and a momentum queue. This yields the set $Y$ used in the InfoNCE term of Equation (16).

We train the model for 400 epochs, including a 5-epoch linear warm-up period, with the batch size of 256. Stochastic gradient descent with an initial learning rate of 0.03, weight decay of 0.0001, and cosine annealing is used for optimization.

We optimize the InfoNCE loss that aligns $f_\theta(P_x)$ with $g_\xi(P_y)$ and repels all $g_\xi(P_y')$ in $Y$ at temperature $\tau=0.07$. The mixed negative construction is important, it prevents shortcuts based on low level statistics and improves invariance to nuisance degradations while preserving sensitivity to content. The MoCo queue of length $K=65{,}536$ provides a stable and diverse negative set across steps.

For the momentum encoder, we set a momentum of 0.999. Data augmentation consists of random cropping with a scaling range of $[0.045, 0.5]$ and an aspect ratio range of $[0.5, 2]$ are applied to both $\{p(\boldsymbol{y}|\boldsymbol{x}_t)\}_{t\in[T]}$ and $\boldsymbol{y}$. For each input image, we crop the given image at a random location, and apply the color jitter as augmentation. For all inverse problems, Gaussian measurement noise with $\sigma = 0.02$ is added (see Appendix L for other noise levels). Full implementation details are available in our code repository.

For the FFHQ, AFHQ and ImageNet datasets, we utilize pretrained score functions following the configuration described in Chung et al. (2022b). To train the score function for kernels, we construct a dataset of 60k $64 \times 64$ kernels. Out of these, 50k motion blur kernels are produced using the implementation from [3], where the blur intensity is sampled as $I \sim \text{Unif}(0.2, 1.0)$. The remaining 10k kernels are Gaussian blurs, generated with a standard deviation chosen randomly as $\sigma \sim \text{Unif}(0.1, 5.0)$.

Similar to the previous works (Chung et al., 2023a; Laroche et al., 2024; Murata et al., 2023; Hamidi & Yang, 2024; Ye et al., 2025), for FFHQ, we randomly select 50k images for training, and sample 1k images of test data separately. For AFHQ, we train our model using the images in the dog category, which consists of about 5k images. Testing was performed with the held-out validation set of 500 images of the same category.

For the kernel and tilt-map score functions, we adopt the U-Net architecture provided in *guided-diffusion* [4], training the models under the default configuration.

## G.2 FORWARD OPERATORS

We define forward operators for rotation blur and zoom blur that we use to synthesize measurements. Let $x \in \mathbb{R}^{H \times W \times C}$ be an image, $c = \left(\frac{H-1}{2}, \frac{W-1}{2}\right)$ the rotation and zoom center, and let $\epsilon \sim \mathcal{N}(0, \sigma_n^2)$ denote measurement noise.

### G.2.1 ROTATION BLUR

Given a maximum shake angle $\phi > 0$, sample angles $\{\theta_i\}_{i=1}^M$ and nonnegative weights $\{w_i\}_{i=1}^M$ that sum to 1. The rotation blur averages rotated views around $c$:

$$y(u) \;=\; \sum_{i=1}^{M} w_i \left[ R_{\theta_i} x \right](u) \;+\; \epsilon(u), \tag{52}$$

where $R_\theta$ applies a rotation of $\theta$ degrees about $c$ with bilinear interpolation and reflect padding.

For our experiments, we set $M = 21$, angles $\theta_i$ linearly spaced in $[-\phi, \phi]$, sample weights $w_i \propto \exp\left(-\frac{\theta_i^2}{2\sigma_\theta^2}\right)$ with $\sigma_\theta = \phi/3$, then normalized so that $\sum_i w_i = 1$. The training range $\phi \sim \text{Unif}(10°, 30°)$. Test ranges follow the benchmark specification. Interpolation is bilinear. Padding is reflect. Channels are processed independently.

---

[3] https://github.com/LeviBorodenko/motionblur
[4] https://github.com/openai/guided-diffusion

### G.2.2  ZOOM BLUR

Given a zoom range $[1, \zeta_{\max}]$, sample scale factors $\{s_i\}_{i=1}^{M}$ and nonnegative weights $\{w_i\}_{i=1}^{M}$ that sum to 1. The zoom blur averages scaled versions about $c$:

$$y(u) = \sum_{i=1}^{M} w_i \left[ Z_{s_i} x \right](u) + \epsilon(u), \tag{53}$$

where $Z_s$ scales by factor $s$ around $c$. For $s > 1$ (minification) we use area interpolation. For $s < 1$ (magnification) we use bilinear interpolation. Padding is reflect.

In the experiments we set $M = 21$, scales $s_i$ linearly spaced in $\left[1, \zeta_{\max}\right]$. Sample weights $w_i \propto \exp\left( - \frac{(s_i - 1)^2}{2\sigma_s^2} \right)$ with $\sigma_s = (\zeta_{\max} - 1)/3$, then normalized. We use the training range of $\zeta_{\max} \sim \text{Unif}(1.1, 1.6)$. Test ranges follow the benchmark specification. Coordinates are centered at $c$. Interpolation choices as above. Unless stated otherwise, we use $\sigma_n = 0.02$ for the additive noise.

### G.3  EVALUATION

We evaluate reconstruction quality using three standard metrics: Fréchet Inception Distance (FID) (Heusel et al., 2017), Learned Perceptual Image Patch Similarity (LPIPS) (Zhang et al., 2018), and Peak Signal-to-Noise Ratio (PSNR). Unless otherwise specified, we fix the number of diffusion steps to 1000 across all experiments.

### G.4  BENCHMARK METHODS

**Pan-$\ell$_0 Pan et al. (2017).** This method applies $\ell$_0 regularization jointly on the image and kernel. We use the official codebase[5] with the following hyper-parameters. Optimization and post-processing follow the same multi-stage strategy as Pan-DCP.

- $\lambda$_pixel $= 4e - 3$
- $\lambda$_grad $= 4e - 3$
- $\lambda$_tv $= 1e - 3$
- $\lambda$_l0 $= 2e - 3$

**SelfDeblur Ren et al. (2020).** We adopt the default YCbCr-based deblurring configuration. Training is performed with a constant learning rate of 0.01 for 2500 iterations. For the first 500 steps, optimization minimizes the MSE loss, after which it switches to minimizing $1 - SSIM(\cdot, \cdot)$.

**DeblurGANv2 Kupyn et al. (2019).** We adopt the official implementation[6], following the default hyper-parameters, data augmentation strategies, and network design. Training minimizes a weighted combination of pixel loss, WGAN-gp adversarial loss, and perceptual loss, with Inception-ResNet-v2 as the generator backbone. Both FFHQ and AFHQ datasets are used, with training conducted for 1.5M iterations using a batch size of 1. As in MPRNet, the training set contains an equal proportion of Gaussian and motion blurred images. The loss weights are:

- $\lambda$_pixel $= 5e - 1$
- $\lambda$_adv $= 6e - 3$
- $\lambda$_perceptual $= 1e - 2$

## H  FURTHER STUDIES ON ENCODER CHOICE

Why `CL-DPS (SPE)` setup is meaningful? In many pipelines a coarse family label is available or can be obtained with negligible cost: (i) capture systems operate in discrete modes that are recorded as metadata, (ii) restoration stacks routinely gate inputs by a small set of operator families before

---

[5]https://jspan.github.io/projects/text-deblurring/index.html
[6]https://github.com/VITA-Group/DeblurGANv2

invoking specialized solvers, (iii) when metadata is missing a lightweight two-way classifier can detect the family with near-perfect accuracy and tiny overhead compared to diffusion sampling. This keeps the setting blind to the unknown continuous parameters while matching practical deployments.

On the other hand, to train `CL-DPS (UNI)`, the learned energy then estimates the *mixture* likelihood $p_{mix}(\boldsymbol{y} \mid \boldsymbol{x}_t)$, which is exactly what is needed when the measurement family is unknown or undetermined at test time. When the family $\mathcal{A}_o$ is known, the same encoder can still be used with a simple restriction of negatives, but no change to the loss is required.

Let $o \in \mathcal{O}$ index the measurement family and $\psi$ denotes family-specific parameters. Given a clean image $\boldsymbol{x}_0$, the forward model produces a measurement

$$\boldsymbol{y} = \mathcal{A}_{o,\psi}(\boldsymbol{x}_0) + n, \tag{54}$$

and let $\boldsymbol{x}_t$ denote a diffusion state along the reverse process. During contrastive learning we first draw the family $o \sim \pi_o$, and then draw $\psi \sim p(\psi \mid O)$, where both $\pi_o$ and $p(\psi \mid O)$ are uniform distributions. We then construct positives $(\boldsymbol{x}_t, \boldsymbol{y})$ from this mixture process, while negatives come from a large dictionary as discussed in Section 4.1.

## H.1 USING HIGHER-CAPACITY ENCODER

To mitigate the performance gap introduced by `CL-DPS (UNI)`, we train a higher-capacity encoder namely ResNet-50, and evaluate it on the FFHQ dataset; results are reported in Table 5. As seen there, scaling up the encoder model largely recovers the degradation caused by the mixture of the operators. We attribute this gain to the greater model capacity of the ResNet-50.

Table 5: `CL-DPS (UNI)` using two model structures, namely ResNet-18 and ResNet-50.

| FFHQ | | | | | | |
|---|---|---|---|---|---|---|
| Encoder | | ResNet-18 | | | ResNet-50 | |
| Distortion | PSNR ↑ | FID ↓ | LPIPS ↓ | PSNR ↑ | FID ↓ | LPIPS ↓ |
| Rotation | 22.27 | 36.55 | 0.315 | 22.60 | 33.95 | 0.309 |
| Zoom | 20.31 | 46.83 | 0.448 | 20.42 | 44.14 | 0.411 |

## H.2 ROBUSTNESS TO FAMILY MISCLASSIFICATION

We simulate a small fraction $\varepsilon$ of inputs routed to the wrong encoder to approximate a practical detector with imperfect accuracy. On FFHQ rotation we vary $\varepsilon \in \{0, 0.05, 0.10, 0.20\}$ and report the PSNR, FID and LPIPS in the Table 6:

Table 6: Numerical results on robustness to family misclassification

| FFHQ | | | |
|---|---|---|---|
| $\varepsilon$ | PSNR | FID | LPIPS |
| 0 | 22.74 | 33.66 | 0.302 |
| 0.05 | 22.57 | 35.35 | 0.310 |
| 0.10 | 22.24 | 36.61 | 0.318 |
| 0.20 | 21.71 | 40.28 | 0.332 |

We observe near-linear degradation for small $\varepsilon$. In our internal test a two-way ResNet-18 family detector reaches $99.1\%$ accuracy on held-out data with a runtime of about $1.1$ ms per $256 \times 256$ image on an Nvidia H100 GPU. Diffusion sampling for 300 steps takes about $5.1$ s per image on the same GPU. The overhead of family detection is therefore negligible relative to sampling time.

Knowing the measurement family yields the best quality. When the family is unknown, a single mixture-trained encoder remains competitive and cross-family usage still outperforms non-contrastive baselines. Small misclassification rates have a modest effect on quality in practice.

# I    DETAILS FOR THE TOY LIKELIHOOD GRADIENT CHECK

To quantify how well the surrogate guidance matches the true likelihood gradient in a case with a closed-form $\nabla_{\boldsymbol{x}_t} \log p(\boldsymbol{y}_t \mid \boldsymbol{x}_t)$, we generate $N_{\text{toy}} = 1100$ grayscale images $\boldsymbol{x}_0 \in [0,1]^{96 \times 96}$ by summing 3 to 6 Gaussian blobs, a few straight edges, and Gaussian noise, then rescaling to $[0,1]$. We use 1000 for training the auxiliary encoder and 100 for diagnostics.

For a parameter $\psi$ we define a linear operator $H_\psi$ applied to the vectorized image. Similar to the experimental setting in Section 5. We sample $\psi$ per image from one of two families:

$$\text{Gaussian blur:} \quad \psi \in [0.6, 2.0], \text{ with kernel size of 5.} \tag{55}$$

$$\text{Motion blur:} \quad \text{intensity of } \in [0.5 - 0.9], \text{ , with kernel size of 5.} \tag{56}$$

At diffusion step $t$ with noise level $\sigma_t$ we draw

$$\boldsymbol{x}_t = \alpha_t \boldsymbol{x}_0 + \sigma_t \boldsymbol{\epsilon}, \quad \boldsymbol{\epsilon} \sim \mathcal{N}(\boldsymbol{0}, I), \tag{57}$$

$$\boldsymbol{y}_t = H_\psi \boldsymbol{x}_t + \boldsymbol{\varepsilon}_t, \quad \boldsymbol{\varepsilon}_t \sim \mathcal{N}(\boldsymbol{0}, \sigma_t^2 I). \tag{58}$$

The true gradient follows

$$g_{\text{true}}(t) = \sigma_t^{-2} H_\psi^\top \big(\boldsymbol{y}_t - H_\psi \boldsymbol{x}_t\big). \tag{59}$$

We train a lightweight encoder $f(\cdot)$ with an InfoNCE loss on pairs $(\boldsymbol{x}_t, \boldsymbol{y}_t)$ drawn from the same forward model. Features are $\ell_2$-normalized. Temperature is $\tau = 0.07$. We use a MoCo queue of $K = 65536$ negatives with momentum $m = 0.996$ and a 128-dimensional projection head. For diagnostics we vary the dictionary size $|Y| \in \{64, 256, 1024\}$ by sampling that many $\boldsymbol{y}_t^{(j)}$ at the same $t$. The surrogate is

$$\widehat{p}_{t,Y}(\boldsymbol{y}_t^{(+)} \mid \boldsymbol{x}_t) = \frac{\exp\big(\langle f(\boldsymbol{x}_t), f(\boldsymbol{y}_t^{(+)})\rangle/\tau\big)}{\sum_j \exp\big(\langle f(\boldsymbol{x}_t), f(\boldsymbol{y}_t^{(j)})\rangle/\tau\big)}. \tag{60}$$

We compute

$$g_{\text{sur}}(t) = \nabla_{\boldsymbol{x}_t} \log \widehat{p}_{t,Y}(\boldsymbol{y}_t^{(+)} \mid \boldsymbol{x}_t) = \nabla s_+ - \sum_j \pi_j \nabla s_j, \quad s_j = \langle f(\boldsymbol{x}_t), f(\boldsymbol{y}_t^{(j)})\rangle/\tau. \tag{61}$$

Main experiments use a 1000-step schedule. Diagnostics use a subsampled grid $\mathcal{T}_{\text{diag}}$ of 50 indices that are equally spaced in $\log \sigma_t$, chosen from the same schedule. This preserves the early, mid, and late regimes with minimal clutter. We use a batch size of 128, projection dimension 32, learning rate 1e$-3$, optimizer AdamW with weight decay 1e$-4$, queue length 16384, momentum 0.996, temperature 0.07. Results are robust to moderate variation of these values.

For each $t \in \mathcal{T}_{\text{diag}}$ and each $|Y|$ we compute

$$\theta_t = \cos^{-1} \frac{\langle g_{\text{true}}(t), g_{\text{sur}}(t)\rangle}{\|g_{\text{true}}(t)\|_2 \|g_{\text{sur}}(t)\|_2}, \qquad \rho_t = \frac{\|g_{\text{sur}}(t)\|_2}{\|g_{\text{true}}(t)\|_2}, \tag{62}$$

averaging over 100 held-out images. We plot the per-t mean with a light moving average for readability. Raw curves and seeds are released with the code.

Under normalized features and a good encoder one expects $\theta_t$ to decrease with $t$ and $\rho_t$ to approach one. Larger $|Y|$ raises the positive softmax weight $\pi_+$ and improves alignment. As a check, increasing $\tau$ or reducing $|Y|$ degrades alignment as expected.

# J    CONVERGENCE BEHAVIOR OF CL-DPS

Here we study convergence under a nonlinear forward model on FFHQ. In Figure 5 we plot PSNR, FID, and LPIPS versus the number of denoising steps. The three baselines from prior work (BlindDPS, FastEM, and GibbsDDRM) do not improve with more steps, and their curves plateau or even drift, indicating that they fail to solve the underlying nonlinear inverse problem. In stark contrast, CL-DPS improves monotonically across all metrics and converges to substantially better reconstructions as the step count increases.

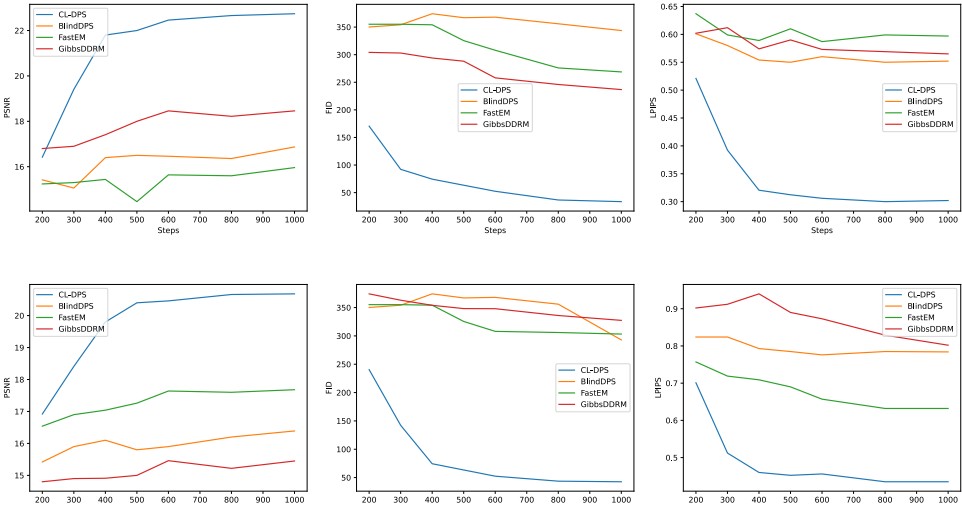

Figure 5: Quantitative evaluation of PSNR, FID, and LPIPS across diffusion steps for challenging nonlinear blind inverse problems. Top: rotation blur; bottom: zoom blur.

Table 7: Wall-clock runtime (per image, in seconds) of CL-DPS compared with diffusion-based baselines.

| Method | FastEM | BlindDPS | LatentEM | GibbsDDRM | CL-DPS |
|---|---|---|---|---|---|
| Time (s) | 46.03 | 47.68 | 46.87 | 51.85 | 90.84 |

## K    RUNTIME AND COMPUTATIONAL OVERHEAD

As discussed in the limitations, CL-DPS has higher runtime because it backpropagates through the encoder at every denoising step. We quantify this overhead by measuring per-image runtime on FFHQ with a single NVIDIA H100 GPU, using the same setting for all methods (1000 sampling steps, batch size 1). Wall-clock times for each method are reported in Table 7.

Under this setup, CL-DPS takes $60.84$ s per image, longer than the baselines in Table 7 due to the extra backpropagations, but this additional cost enables CL-DPS to solve the non-linear inverse problems evaluated in our experiments, where the compared methods fail. In short, CL-DPS trades a modest increase in wall-clock time for substantially broader applicability.

## L    ADDITIONAL RESULTS ACROSS DIFFERENT NOISE LEVELS

To evaluate the robustness of CL-DPS under varying levels of measurement noise, we report additional results for both **linear** and **nonlinear** blind inverse problems across three Gaussian noise levels: $\sigma = 0.01$, $\sigma = 0.02$, and $\sigma = 0.03$.

Table 8 and Table 1 (main paper) and Table 9 present results for blind rotation and zoom deblurring on FFHQ and AFHQ datasets under $\sigma = 0.01$, $0.02$, and $0.03$, respectively. As expected, increasing the noise level degrades the performance of all methods across PSNR, FID, and LPIPS. Nonetheless, CL-DPS consistently outperforms all baselines in both distortion and perceptual metrics, particularly in high-noise and nonlinear settings where competing methods struggle.

Similarly, to evaluate the linear blind inverse problems, we evaluate blind motion and Gaussian deblurring tasks under varying noise levels in Table 10 ($\sigma = 0.01$), Table 2 (main paper, $\sigma = 0.02$), and Table 11 ($\sigma = 0.03$). CL-DPS again delivers competitive or superior performance across all metrics. Notably, CL-DPS maintains high reconstruction quality even at $\sigma = 0.03$, while other methods suffer significant degradation. This highlights the robustness and generalization ability of our contrastive-guided posterior sampling framework under blind measurement noise.

Table 8: **Nonlinear** blind inverse problems at $\sigma = 0.01$. Bold and underlined values denote best and second-best, respectively.

| Method | FFHQ ($256 \times 256$) | | | | | | AFHQ ($256 \times 256$) | | | | | |
| | Rotation | | | Zoom | | | Rotation | | | Zoom | | |
| | PSNR ↑ | FID ↓ | LPIPS ↓ | PSNR ↑ | FID ↓ | LPIPS ↓ | PSNR ↑ | FID ↓ | LPIPS ↓ | PSNR ↑ | FID ↓ | LPIPS ↓ |
|---|---|---|---|---|---|---|---|---|---|---|---|---|
| CL-DPS (SPE) | **24.35** | **28.93** | **0.274** | **22.12** | **38.21** | **0.412** | **22.85** | **33.50** | **0.295** | **21.05** | **53.45** | **0.445** |
| BlindDPS | 17.50 | 310.2 | 0.540 | 17.11 | 260.1 | 0.765 | 14.48 | 185.4 | 0.651 | 12.54 | 250.5 | 0.585 |
| FastEM | 16.61 | 240.1 | 0.588 | 19.82 | 280.1 | 0.615 | 12.43 | 260.1 | 0.663 | 16.63 | 290.3 | 0.786 |
| GibbsDDRM | 19.51 | 215.1 | 0.552 | 16.51 | 300.24 | 0.781 | 16.34 | 240.4 | 0.611 | 15.63 | 260.19 | 0.534 |

Table 9: **Nonlinear** blind inverse problems at $\sigma = 0.03$. Bold and underlined values denote best and second-best, respectively.

| Method | FFHQ ($256 \times 256$) | | | | | | AFHQ ($256 \times 256$) | | | | | |
| | Rotation | | | Zoom | | | Rotation | | | Zoom | | |
| | PSNR ↑ | FID ↓ | LPIPS ↓ | PSNR ↑ | FID ↓ | LPIPS ↓ | PSNR ↑ | FID ↓ | LPIPS ↓ | PSNR ↑ | FID ↓ | LPIPS ↓ |
|---|---|---|---|---|---|---|---|---|---|---|---|---|
| CL-DPS (SPE) | **21.22** | **38.50** | **0.332** | **19.28** | **48.36** | **0.461** | **19.61** | **42.53** | **0.345** | **18.17** | **63.67** | **0.542** |
| BlindDPS | 16.31 | 365.4 | 0.568 | 15.54 | 320.5 | 0.867 | 12.75 | 215.4 | 0.691 | 11.25 | 290.1 | 0.634 |
| FastEM | 15.23 | 290.2 | 0.615 | 17.81 | 326.0 | 0.647 | 11.27 | 310.0 | 0.749 | 15.10 | 335.7 | 0.821 |
| GibbsDDRM | 17.57 | 260.0 | 0.584 | 14.52 | 340.3 | 0.820 | 14.53 | 295.8 | 0.661 | 13.70 | 310.12 | 0.561 |

Table 10: **Linear** blind inverse problems at $\sigma = 0.01$. Bold and underlined values denote best and second-best, respectively.

| Method | FFHQ ($256 \times 256$) | | | | | | AFHQ ($256 \times 256$) | | | | | |
| | Motion | | | Gaussian | | | Motion | | | Gaussian | | |
| | PSNR↑ | FID↓ | LPIPS↓ | PSNR↑ | FID↓ | LPIPS↓ | PSNR↑ | FID↓ | LPIPS↓ | PSNR↑ | FID↓ | LPIPS↓ |
|---|---|---|---|---|---|---|---|---|---|---|---|---|
| CL-DPS (SPE) | **27.49** | 28.54 | 0.142 | **26.21** | **23.84** | 0.325 | **23.57** | 28.99 | 0.185 | **25.27** | **18.34** | **0.216** |
| SelfDeblur | 12.82 | 236.1 | 0.732 | 13.31 | 210.4 | 0.667 | 10.73 | 270.4 | 0.743 | 13.44 | 152.54 | 0.642 |
| DeblurGANv2 | 19.33 | 180.2 | 0.541 | 21.14 | 155.3 | 0.511 | 18.75 | 155.6 | 0.565 | 22.25 | 74.24 | 0.495 |
| Pan-$\ell_0$ | 18.11 | 82.35 | 0.385 | 22.22 | 75.76 | 0.361 | 17.88 | 205.2 | 0.603 | 23.92 | 56.35 | 0.365 |
| BlindDPS | 22.95 | **26.51** | 0.270 | 25.63 | 24.82 | **0.226** | 21.64 | **22.19** | 0.320 | 24.52 | 18.82 | 0.275 |
| FastEM | 25.01 | 31.24 | 0.327 | 23.82 | 27.95 | 0.352 | 22.12 | 46.84 | 0.394 | 23.34 | 29.53 | 0.287 |
| LatentDEM | 24.42 | 34.84 | 0.155 | 25.57 | 31.31 | 0.345 | 21.22 | 41.52 | 0.273 | 22.46 | 35.87 | 0.275 |
| GibbsDDRM | 26.65 | 36.25 | **0.150** | 27.34 | 30.85 | 0.410 | 22.96 | 44.24 | **0.172** | 24.43 | 39.23 | 0.384 |

Table 11: **Linear** blind inverse problems at $\sigma = 0.03$. Bold and underlined values denote best and second-best, respectively.

| Method | FFHQ ($256 \times 256$) | | | | | | AFHQ ($256 \times 256$) | | | | | |
| | Motion | | | Gaussian | | | Motion | | | Gaussian | | |
| | PSNR↑ | FID↓ | LPIPS↓ | PSNR↑ | FID↓ | LPIPS↓ | PSNR↑ | FID↓ | LPIPS↓ | PSNR↑ | FID↓ | LPIPS↓ |
|---|---|---|---|---|---|---|---|---|---|---|---|---|
| CL-DPS (SPE) | **24.36** | 38.26 | 0.146 | **23.03** | **31.06** | 0.373 | **20.81** | 38.88 | 0.235 | **22.62** | **24.54** | **0.255** |
| SelfDeblur | 9.87 | 304.1 | 0.730 | 10.47 | 260.5 | 0.695 | 8.37 | 330.3 | 0.790 | 10.53 | 190.43 | 0.673 |
| DeblurGANv2 | 16.94 | 235.5 | 0.600 | 18.36 | 204.3 | 0.555 | 16.83 | 207.1 | 0.625 | 19.45 | 95.64 | 0.545 |
| Pan-$\ell_0$ | 14.93 | 260.5 | 0.560 | 19.22 | 115.8 | 0.430 | 14.65 | 255.8 | 0.645 | 20.03 | 75.73 | 0.421 |
| BlindDPS | 21.52 | **34.53** | 0.304 | 23.84 | 32.42 | **0.251** | 19.92 | **29.63** | 0.360 | 22.64 | 23.54 | 0.310 |
| FastEM | 23.66 | 42.20 | 0.365 | 22.16 | 37.52 | 0.390 | 20.84 | 62.04 | 0.335 | 22.63 | 40.45 | 0.317 |
| LatentDEM | 21.97 | 45.76 | 0.175 | 23.86 | 40.45 | 0.385 | 19.79 | 54.47 | 0.302 | 21.27 | 42.54 | 0.305 |
| GibbsDDRM | 24.58 | 46.55 | **0.132** | 25.45 | 41.54 | 0.455 | 21.40 | 55.43 | **0.215** | 22.83 | 51.73 | 0.335 |

# M  ABLATION STUDY

## M.1  ABLATION ON CONTRASTIVE HYPERPARAMETERS

We study sensitivity to the InfoNCE temperature $\tau$, the dictionary size $|Y|$ used at guidance time, the MoCo queue length $K$ during pretraining, the projection head dimension $d$, patch size $P$, and stride $S$.

The default parameters used in the experiments in the main body of the paper are $\tau = 0.07$, $|Y| = 256$, $K = 65536$, momentum $m = 0.996$, $d = 128$, $P = 64$ and $S = 32$. In the following, we perform ablation over all these hyperparameters. Reported numbers are averaged over Rotation and Zoom on the three benchmarks used in Table 1.

**Ablation on $\tau$, Table 12.**  We observe that CL-DPS is relatively robust to the temperature $\tau$ in the range $0.05 \leq \tau \leq 0.10$. A slightly lower temperature ($\tau = 0.05$) improves PSNR and FID on FFHQ and AFHQ, though it slightly worsens ImageNet fidelity (FID rises by 0.5). Larger $\tau$ values

Table 12: Sensitivity to temperature $\tau$ in the InfoNCE loss for `CL-DPS (SPE)` setup.

| $\tau$ | FFHQ | | | AFHQ | | | ImageNet | | |
|---|---|---|---|---|---|---|---|---|---|
| | PSNR↑ | FID↓ | LPIPS↓ | PSNR↑ | FID↓ | LPIPS↓ | PSNR↑ | FID↓ | LPIPS↓ |
| 0.05 | 22.88 | 33.12 | 0.298 | 21.61 | 36.43 | 0.316 | 20.38 | 45.64 | 0.344 |
| 0.07 (default) | 22.74 | 33.66 | 0.302 | 21.46 | 36.96 | 0.319 | 20.45 | 45.10 | 0.340 |
| 0.10 | 22.67 | 34.25 | 0.306 | 21.32 | 37.55 | 0.323 | 20.36 | 46.02 | 0.345 |
| 0.15 | 22.41 | 35.11 | 0.312 | 21.15 | 38.49 | 0.329 | 20.12 | 46.98 | 0.352 |

($\tau = 0.15$) consistently degrade quality across all datasets, confirming that overly smooth logits reduce the effectiveness of the contrastive guidance signal.

Table 13: Sensitivity to dictionary size $|Y|$ used during guidance in `CL-DPS (SPE)` setup (for the denominator-aware version only).

| $|Y|$ | FFHQ | | | AFHQ | | | ImageNet | | |
|---|---|---|---|---|---|---|---|---|---|
| | PSNR↑ | FID↓ | LPIPS↓ | PSNR↑ | FID↓ | LPIPS↓ | PSNR↑ | FID↓ | LPIPS↓ |
| 64 | 22.45 | 35.21 | 0.312 | 21.20 | 38.12 | 0.329 | 20.18 | 46.81 | 0.352 |
| 256 (default) | 22.74 | 33.66 | 0.302 | 21.46 | 36.96 | 0.319 | 20.05 | 45.10 | 0.340 |
| 1024 | 22.59 | 32.79 | 0.296 | 21.66 | 35.89 | 0.313 | 20.62 | 44.17 | 0.337 |

**Ablation on $|Y|$, Table 13.** Increasing the dictionary size $|Y|$ improves all three metrics, with the largest gain seen when moving from $|Y| = 64$ to $|Y| = 256$. Gains from $|Y| = 256$ to $|Y| = 1024$ are smaller, suggesting diminishing returns. The trend is most pronounced on ImageNet, where FID improves by more than one point, indicating that a richer dictionary better approximates the true likelihood gradient in high-diversity datasets.

Table 14: Sensitivity to MoCo queue length $K$ for `CL-DPS (SPE)` setup.

| $K$ | FFHQ | | | AFHQ | | | ImageNet | | |
|---|---|---|---|---|---|---|---|---|---|
| | PSNR↑ | FID↓ | LPIPS↓ | PSNR↑ | FID↓ | LPIPS↓ | PSNR↑ | FID↓ | LPIPS↓ |
| 8192 | 22.53 | 34.84 | 0.308 | 21.28 | 37.85 | 0.325 | 20.26 | 46.24 | 0.347 |
| 16384 | 22.61 | 34.33 | 0.306 | 21.36 | 37.28 | 0.322 | 20.33 | 45.83 | 0.343 |
| 65536 (default) | 22.74 | 33.66 | 0.302 | 21.46 | 36.96 | 0.319 | 20.05 | 45.10 | 0.340 |
| 131072 | 22.69 | 33.59 | 0.321 | 21.56 | 36.42 | 0.317 | 20.42 | 44.81 | 0.338 |

**Ablation on $K$, Table 14.** Queue length $K$ plays a similar role to dictionary size by providing harder negatives during pretraining. We observe consistent improvements as $K$ grows, with the most notable jump between $K = 8192$ and $K = 65536$. The improvement saturates beyond $K = 65536$, where increasing to $K = 131072$ yields only marginal additional benefit while incurring higher memory cost.

**Ablation on $d$, Table 15.** The projection dimension $d$ also affects representation quality. Larger dimensions yield modest improvements across all benchmarks, but the relative gain between $d = 128$ and $d = 256$ is small compared to the additional computation and memory footprint. Hence, $d = 128$ offers a good trade-off between performance and efficiency for our default configuration. We show the visualization results in Figure 7.

**Ablation on $P$ & $S$, Table 16.** This table demonstrates the effect of patch size $P$ and stride $S$ on CL-DPS performance for both rotation and zoom blur. Reducing the stride from $S$=32 to $S$=16 (increasing overlap) consistently improves PSNR, lowers FID, and reduces perceptual error across all datasets, confirming that denser spatial coverage produces more stable guidance. Using a smaller patch size ($P$=48) provides similar but slightly weaker gains, indicating that context loss offsets some benefits of denser sampling. Larger patches ($P$=96) or no overlap ($S$=64) consistently

Table 15: Sensitivity to projection head dimension $d$ for `CL-DPS (SPE)` setup.

| $d$ | FFHQ | | | AFHQ | | | ImageNet | | |
|---|---|---|---|---|---|---|---|---|---|
| | PSNR↑ | FID↓ | LPIPS↓ | PSNR↑ | FID↓ | LPIPS↓ | PSNR↑ | FID↓ | LPIPS↓ |
| 64 | 22.55 | 34.51 | 0.307 | 21.56 | 37.51 | 0.324 | 20.28 | 45.94 | 0.346 |
| 128 (default) | 22.74 | 33.66 | 0.302 | 21.46 | 36.96 | 0.319 | 20.45 | 45.10 | 0.340 |
| 256 | 22.75 | 33.69 | 0.299 | 21.58 | 36.89 | 0.316 | 20.55 | 45.17 | 0.339 |

Table 16: Sensitivity to patch size and stride. Default CL-DPS uses patch size $P=64$ and stride $S=32$ (50% overlap). Increasing overlap (smaller stride) consistently helps; removing overlap ($S=64$) hurts. We use `CL-DPS (SPE)` setup.

| | FFHQ ($256 \times 256$) | | | AFHQ ($256 \times 256$) | | | ImageNet | | |
|---|---|---|---|---|---|---|---|---|---|
| Rotation | PSNR↑ | FID↓ | LPIPS↓ | PSNR↑ | FID↓ | LPIPS↓ | PSNR↑ | FID↓ | LPIPS↓ |
| CL-DPS (P=64, S=32) | 22.74 | 33.66 | 0.302 | 21.46 | 36.96 | 0.319 | 20.05 | 45.06 | 0.342 |
| CL-DPS (P=64, S=16) | 23.09 | 32.10 | 0.292 | 21.78 | 35.50 | 0.312 | 20.73 | 43.83 | 0.332 |
| CL-DPS (P=48, S=16) | 22.98 | 32.61 | 0.296 | 21.65 | 35.90 | 0.315 | 20.66 | 44.22 | 0.334 |
| CL-DPS (P=96, S=48) | 22.25 | 35.13 | 0.314 | 21.15 | 38.07 | 0.337 | 20.21 | 46.81 | 0.353 |
| CL-DPS (P=64, S=64) | 21.94 | 36.90 | 0.330 | 20.68 | 40.23 | 0.344 | 19.70 | 49.74 | 0.368 |

| | FFHQ ($256 \times 256$) | | | AFHQ ($256 \times 256$) | | | ImageNet | | |
|---|---|---|---|---|---|---|---|---|---|
| Zoom | PSNR↑ | FID↓ | LPIPS↓ | PSNR↑ | FID↓ | LPIPS↓ | PSNR↑ | FID↓ | LPIPS↓ |
| CL-DPS (P=64, S=32) | 20.68 | 42.61 | 0.435 | 19.63 | 57.54 | 0.468 | 18.56 | 55.30 | 0.481 |
| CL-DPS (P=64, S=16) | 21.12 | 40.21 | 0.422 | 20.15 | 54.34 | 0.451 | 18.95 | 52.90 | 0.466 |
| CL-DPS (P=48, S=16) | 21.05 | 40.80 | 0.424 | 20.02 | 55.03 | 0.458 | 18.86 | 53.80 | 0.472 |
| CL-DPS (P=96, S=48) | 20.32 | 44.56 | 0.444 | 19.34 | 58.67 | 0.482 | 18.35 | 56.80 | 0.498 |
| CL-DPS (P=64, S=64) | 19.78 | 47.30 | 0.462 | 18.72 | 61.51 | 0.496 | 17.71 | 59.93 | 0.506 |

hurt reconstruction quality, especially on ImageNet, suggesting that overly coarse or disjoint patch coverage fails to capture sufficient local detail.

We also provide an example in Figure 6 to show how the patchification works.

In addition, in order to show that how $P$ & $S$ values affect the quality of the restored images, we plot the images reconstructed by CL-DPS using different $P$ & $S$ values. The results are reported in Figure 7.

## M.2 ABLATION ON COLOR CONSISTENCY HEAD

We qualitatively evaluate the effect of the color consistency head on reconstruction quality. As shown in Figure 8, the model trained *without* the color consistency head (Figure 8c) produces a restored image with noticeable color shifts, particularly in the shirt region. In contrast, the model trained *with* the color consistency head (Figure 8d) recovers colors faithfully, resulting in a visually consistent reconstruction that better matches the original image. This confirms that incorporating the color consistency head stabilizes the color distribution during training and prevents hue drift in the restored outputs.

## N QUALITATIVE RESULTS ON ZOOM DEBLURRING

Zoom blur is among the most challenging nonlinear degradations for diffusion-based inverse solvers, often causing existing methods to produce severe artifacts or completely fail. As illustrated in Figure 12, benchmark methods such as BlindDPS, FastEM, and GibbsDDRM struggle to recover fine details and exhibit strong distortions. In contrast, CL-DPS reconstructs a visually coherent image with accurate structure and color, demonstrating its robustness under extreme nonlinear conditions. These results highlight that CL-DPS is not merely competitive but uniquely capable of handling severe zoom blur without catastrophic failure.

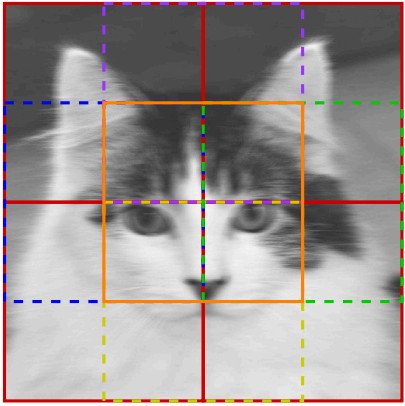

Figure 6: Example of patchified image of a resolution $256 \times 256$, with a stride size of $64$ and a patch size of $128 \times 128$.

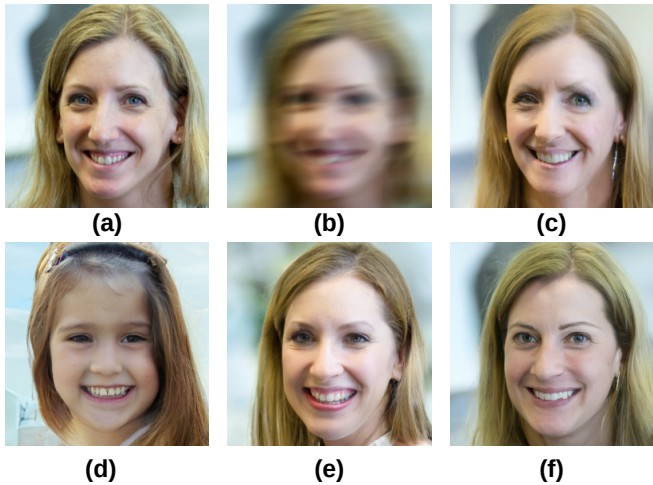

Figure 7: Ablation study of different inference strategies. (a) Original image, (b) Measurement, (c) Restored image with stride 64, patch size 32, (d) Stride 256, patch size 256, (e) Stride 64, patch size 64, (f) Stride 48, patch size 16.

## O   QUALITATIVE RESULTS ON LINEAR DEBLURRING TASK

Gaussian blur and motion blur are two well-studied cases in blind diffusion-based deblurring. To demonstrate CL-DPS's superiority on the blind linear deblurring task, we present additional visual results and compare reconstructions against two baselines, BlindDPS (Chung et al., 2023a) and

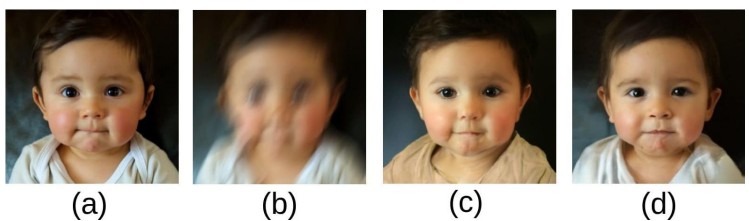

Figure 8: Ablation study on the color consistency head. (a) Original image, (b) measurement, (c) restoration without the color consistency head (color shift visible), (d) restoration with the color consistency head (color faithfully preserved).

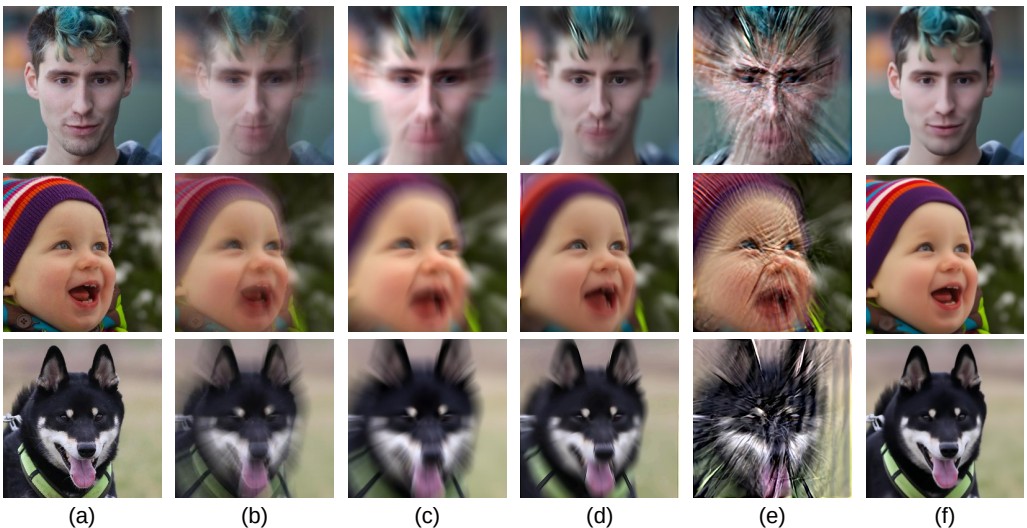

Figure 9: Qualitative results on blind zoom deblurring, a challenging nonlinear inverse problem. (a) Ground truth image, (b) zoom-blurred measurement, and restorations from (c) BlindDPS (Chung et al., 2023a), (d) FastEM (Laroche et al., 2024), (e) GibbsDDRM (Murata et al., 2023), and (f) CL-DPS (ours). Competing methods fail catastrophically, whereas CL-DPS successfully reconstructs a sharp and color-consistent image.

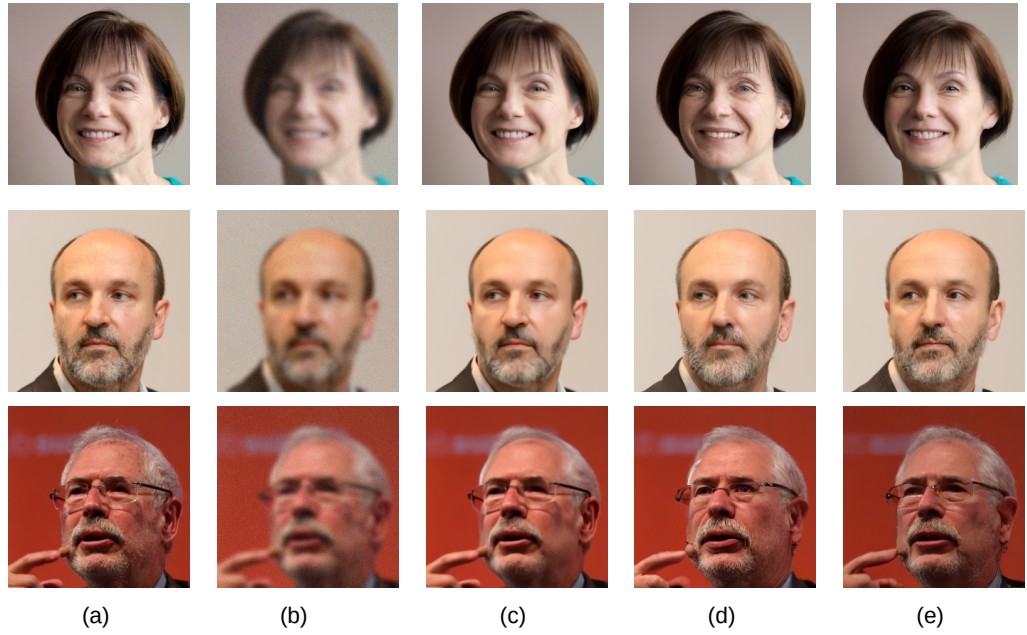

Figure 10: Blind Gaussian deblurring results: (a) ground-truth image, (b) Gaussian-blurred measurement, and restorations using (c) BlindDPS (Chung et al., 2023a), (d) GibbsDDRM (Murata et al., 2023), and (e) CL-DPS (ours). Visually, CL-DPS produces more natural images.

GibbsDDRM (Murata et al., 2023). Results for Gaussian blur and motion blur are shown in Figure 10 and Figure 11, respectively.

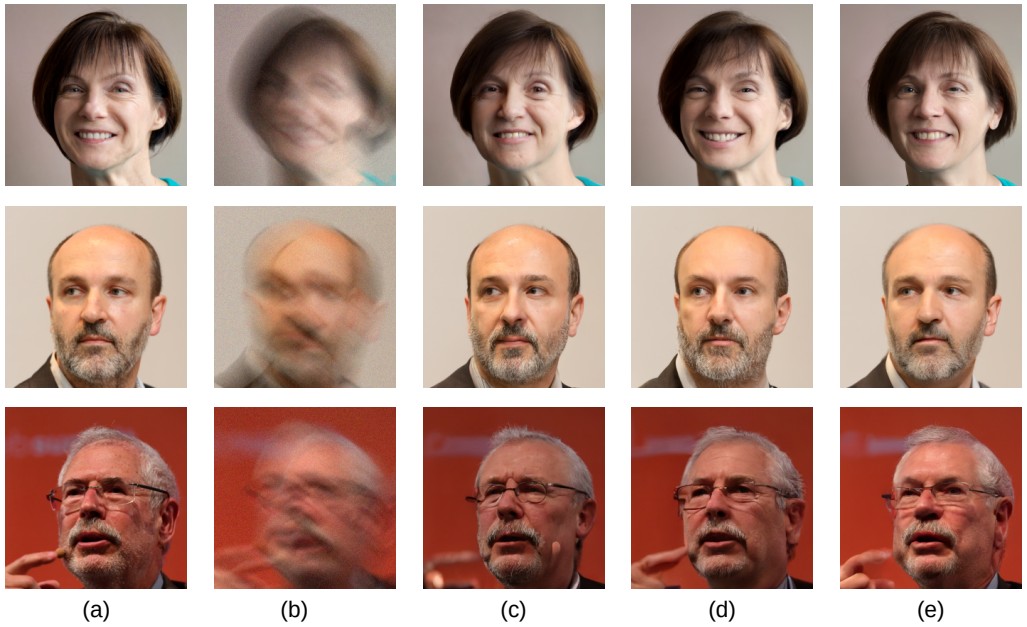

Figure 11: Blind motion deblurring results: (a) ground-truth image, (b) motion-blurred measurement, and restorations using (c) BlindDPS (Chung et al., 2023a), (d) GibbsDDRM (Murata et al., 2023), and (e) CL-DPS (ours). Visually, CL-DPS produces more natural images.

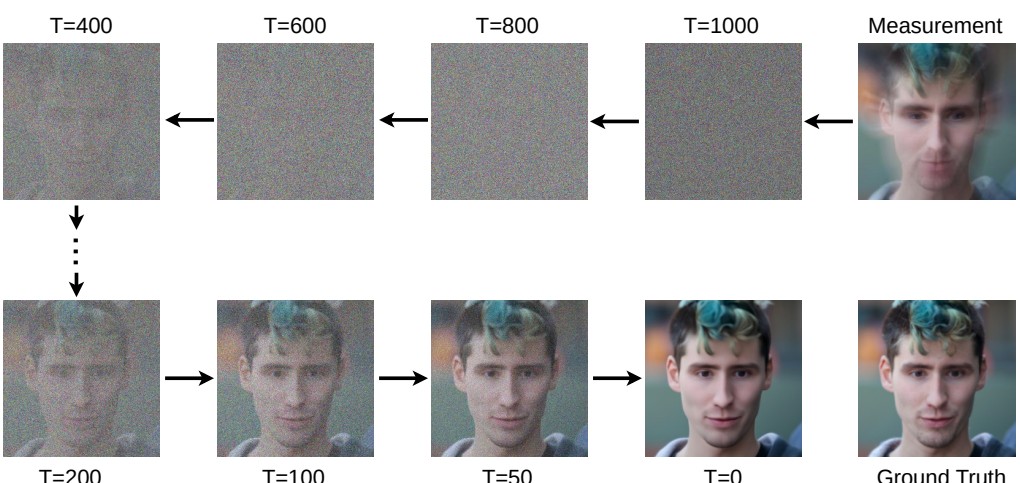

Figure 12: The CL-DPS process of recovering the zoom blurred measurement.

## P  DENOISING PROCESS OF CL-DPS

Here, we visualize the denoising process of CL-DPS over 1000 timesteps. To this end, we select a single image and display the reconstructed images throughout the denoising process, as illustrated in Figure 12.

