# OpenReview forum: "CL-DPS: A Contrastive Learning Approach to Blind Nonlinear Inverse Problem Solving via Diffusion Posterior Sampling"
_ICLR.cc/2026/Conference — ICLR 2026 Poster_

### Official Review · Reviewer_qhCf · 2025-10-29

**Soundness:** 3
**Presentation:** 3
**Contribution:** 3
**Rating:** 8
**Confidence:** 4

**Summary:**

Posterior sampling with diffusion models requires computing the gradient term $\nabla_{x_t} \log p(y \mid x_t)$ when solving the reverse ODE. To estimate this term, the paper trains an encoder $f$ via contrastive learning, such that the log-likelihood gradient can be approximated by the gradient of a contrastive score - defined as the inner product between $f(x_t)$ and $f(y)$. By training the encoder on synthetic measurements, the method enables blind inverse problem solving for both linear and nonlinear forward operators. Extensive experiments on Gaussian, motion, rotational, and zoom deblurring tasks demonstrate that the proposed approach outperforms existing baseline models in reconstruction quality.

**Strengths:**

- The paper proposes a way to solve non-linear blind inverse problem, which is not actively explored with existing baseline method, yet.
- The paper provides various empirical analysis on effects of introduced components (e.g. patch-wise inference, regularization, dictionary size...) as well as theoretical analysis.
- The proposed method is more effective than baseline methods.

**Weaknesses:**

- Problem is partially blind: encoders are trained for a family of measurement, but with limited number of factors (e.g. two known angles for rotation blur).
- Missing details for the forward operations. Especially, there is not sufficient information for implementing zoom and rotation blur.
- Missing related works [1], which is originally proposed for linear inverse problem, but could be used for non-linear inverse problem too.


Reference

[1] Diffusion Prior-Based Amortized Variational Inference for Noisy Inverse Problems, ECCV2024

**Questions:**

How the authors applied BlindDPS for zoom and rotation blur? As diffusion prior for blur kernel is only trained for the gaussian and motion blur, it is not straight-forward to use it for those tasks.

---

> ### Author Response · Authors · 2025-11-23
>
> Thank you for the very positive evaluation. We are glad the reviewer found the problem setting, analyses, and empirical results compelling. We will keep polishing clarity, highlight the ablations on patch overlap and dictionary size, and expand any details that would further strengthen reproducibility.
>
> **Weaknesses:**
> >1. Problem is partially blind: encoders are trained for a family of measurement, but with limited number of factors (e.g. two known angles for rotation blur).
>
> We appreciate your attention to this issue. We believe this concern arises from a misunderstanding, which we have addressed in the revision. We do not train on a limited set of fixed factors. Instead, during encoder training, measurement parameters are continuously sampled from broad ranges for each minibatch. As a result, the encoder encounters a distribution of operators rather than a small set of pre-specified angles or zoom levels. Additionally, CL-DPS operates without access to operator information at test time. In the Universal (UNI) setting, a single encoder is trained on a mixture of operator families and is then applied at test time without access to operator parameters, operator labels, or any routing between specialist models.
>
> We will explicitly clarify these points in the revised manuscript to prevent any misunderstanding that the encoder is trained solely on “two known angles” or a similarly limited set of factors.
>
> > 2. Missing details for the forward operations. Especially, there is not sufficient information for implementing zoom and rotation blur.
>
> Thank you for flagging this. We have revised Appendix G2 to include the full detail on implementing zoom and rotation blur.
>
> > 3. Missing related works [1], which is originally proposed for linear inverse problem, but could be used for non-linear inverse problem too.
>
> Thank you for highlighting reference [1]. We have updated Section 2 (Related Work) to include a concise discussion of [1]. Conceptually, [1] implements amortized variational inference with a diffusion prior, typically applied to known forward models. In contrast, CL-DPS addresses the blind nonlinear setting: we maintain a fixed diffusion prior, do not assume or estimate operator parameters, and introduce a contrastive likelihood surrogate as plug-and-play guidance. Empirically, this approach enables us to address rotation and zoom blur without requiring operator access.
>
> [1] Diffusion Prior-Based Amortized Variational Inference for Noisy Inverse Problems, ECCV2024
>
> **Questions:**
>
> >How the authors applied BlindDPS for zoom and rotation blur? As diffusion prior for blur kernel is only trained for the gaussian and motion blur, it is not straight-forward to use it for those tasks.
>
> Adapting BlindDPS to rotation/zoom would require designing and training a new **nonlinear operator prior** and re-deriving its likelihood, which constitutes a **new baseline variant**, not the published method. To keep comparisons faithful, we evaluate methods **as released**, document configs, and use identical priors, datasets, and step counts across methods.

---

### Official Review · Reviewer_c2Zw · 2025-10-30

**Soundness:** 4
**Presentation:** 3
**Contribution:** 3
**Rating:** 6
**Confidence:** 3

**Summary:**

This paper introduces a framework for solving blind nonlinear inverse problems where existing diffusion models fail. It works by training an auxiliary contrastive encoder offline to learn a likelihood surrogate, which is then used as gradient guidance during diffusion posterior sampling to restore images without knowing the specific degradation operator. The evaluations are comprehensive.

**Strengths:**

1. The paper manages to apply diffusion models to the blind nonlinear inverse problem, a domain where previous SOTA DM-based methods often fail.

2. The core idea of using an offline, MoCo-trained auxiliary encoder to learn an amortized likelihood surrogate over a distribution of operators is reasonable and novel.

3. The empirical results are comprehensive. CL-DPS produces high-quality restorations on challenging nonlinear tasks (rotational/zoom deblurring).

**Weaknesses:**

1. Inference Cost: The primary weakness is computational overhead. The method requires a forward and backward pass through the auxiliary encoder at every sampling step to compute the guidance gradient. This is a significant practical limitation. Can the cost be further reduced?

2. Reliance on Special Encoders: The best-performing model requires training a separate encoder for each family of operators. This assumes the operator class is known at test time, which is a strong assumption that weakens the "fully blind" claim. The fully blind UNI model shows a consistent performance drop. While Appendix H.1 shows that a larger ResNet-50 can close this gap, this trade-off between generality and performance/efficiency is a key weakness.

3. Mismatch in Guidance: The main algorithm uses a simplified "numerator-only" guidance gradient, which deviates from the full, theoretically-derived gradient. While Appendix E provides an empirical justification (showing the denominator term is small and its inclusion gives minimal benefit for high cost), this is a disconnect between the presented theory and the practical implementation.

4. How does the model perform on OOD parameters, such as a $40^{\circ}$ rotation blur? Does it degrade gracefully or fail abruptly?

**Questions:**

See weaknesses.

---

> ### Author Response · Authors · 2025-11-23
>
> Thank you for the thoughtful and encouraging assessment. We appreciate your recognition of the novelty, scope, and empirical depth of CL-DPS.
>
> Please find our concise responses to your specific concerns below.
>
> **Weaknesses:**
>
> >1.  Inference Cost: The primary weakness is computational overhead. The method requires a forward and backward pass through the auxiliary encoder at every sampling step to compute the guidance gradient. This is a significant practical limitation. Can the cost be further reduced?
>
> Thank you for highlighting the runtime. CL-DPS indeed requires one forward and one backward pass through a lightweight auxiliary encoder at each diffusion step to compute the guidance gradient. In practice, however, this overhead is moderate: as reported in Table 7, CL-DPS takes approximately 20% more time to recover one image under the same diffusion prior, number of sampling steps, and hardware setup. Given that, to the best of our knowledge, CL-DPS is the first diffusion-based method capable of solving blind nonlinear inverse problems, we view this additional cost as a favorable trade-off for the substantially expanded problem class it can handle.
>
> Regarding cost reduction, the main bottleneck is the backward pass through the auxiliary encoder. A natural way to decrease runtime is therefore to design more efficient encoder architectures tailored to this task. We have revised the Conclusion and Future Work section to explicitly discuss this direction as a promising avenue for further improving efficiency without changing the core algorithmic framework.
>
> > 2.  Reliance on Special Encoders: The best-performing model requires training a separate encoder for each family of operators. This assumes the operator class is known at test time, which is a strong assumption that weakens the "fully blind" claim. The fully blind UNI model shows a consistent performance drop. While Appendix H.1 shows that a larger ResNet-50 can close this gap, this trade-off between generality and performance/efficiency is a key weakness.
>
> Thank you for highlighting this trade-off. UNI operates in a fully blind manner and outperforms all baselines. Across all operator families, including nonlinear rotation and zoom scenarios, the single universal encoder (UNI), trained once on a mixture of operators, consistently outperforms all baseline diffusion methods. The performance difference between UNI and SPE is relatively minor, while the gap between UNI and the strongest baselines is considerably larger. Therefore, even without operator information at test time, UNI maintains a clear advantage. Furthermore, as shown in Appendix H.1, a modest increase in model capacity, such as employing a ResNet-50 encoder, recovers most of the UNI-to-SPE gap at minimal cost, without requiring operator labels or routing.
>
> > 3. Mismatch in Guidance: The main algorithm uses a simplified "numerator-only" guidance gradient, which deviates from the full, theoretically-derived gradient. While Appendix E provides an empirical justification (showing the denominator term is small and its inclusion gives minimal benefit for high cost), this is a disconnect between the presented theory and the practical implementation.
>
> Thank you for pointing this out. Analogous approximations are standard in energy-based models and contrastive learning, where gradients are taken with respect to unnormalized scores and partition/denominator terms are treated as constants or estimated only coarsely. Our choice to omit the denominator during guidance is thus a pragmatic instance of this common pattern: we retain the dominant, task-aligned energy term while avoiding an expensive, low-signal correction.
>
> Theoretically,  we use an energy $E(x_t, y)$ as a surrogate log-likelihood. The guidance is $\nabla_{x_t}\log p(y \mid x_t) \propto \nabla_{x_t} E(x_t, y)$. The partition term $\nabla_{x_t}\log Z(y)$ is zero because $Z$ integrates over a dummy variable $x'$ and does not depend on $x_t$. The practical “full” MoCo form introduces a denominator over **negatives tied to the current $x_t$**, which yields an extra term that subtracts a weighted average of negative similarities. Our “numerator-only” guidance simply drops this **contrastive** correction at inference. Empirically,  negatives are numerous and approximately isotropic due to the large queue and hard-negative mix, so their weighted average is near zero. In **Appendix E**, we report that the denominator term’s gradient norm is typically **5–8 percent** of the numerators across timesteps and operator families.

---

> ### Author Response · Authors · 2025-11-23
>
> **Weakness:**
>
> >4. How does the model perform on OOD parameters, such as a 40 degree rotation blur? Does it degrade gracefully or fail abruptly?
>
> We agree that understanding the behavior under OOD operator parameters is important. To directly address this, we added a new experiment on the Rotation Blur task. The CL-DPS model demonstrates graceful degradation in OOD parameters, such as a 40-degree rotation blur.
>
> We specifically performed an experiment to test extrapolation on the Rotation Blur task, where the standard In-Distribution training range was [10°, 30°] and we tested the method against [30°, 40°]. The results are listed in the following table:
>
> | Model | In-range PSNR | OOD PSNR | Δ |
> |---------|-------|-------|-------|
> | CL-DPS  | 22.74 | 19.86 | −2.88 dB |

---

### Official Review · Reviewer_unhN · 2025-11-01

**Soundness:** 3
**Presentation:** 3
**Contribution:** 3
**Rating:** 6
**Confidence:** 4

**Summary:**

This paper proposes a diffusion-based framework for solving nonlinear blind inverse problems, a setting that remains largely underexplored in the current literature.
The authors introduce a MoCo-style auxiliary encoder, which is ingeniously integrated into the diffusion inference process to address the challenges brought by both nonlinearity and blindness in the observation model.
Experimental results demonstrate that the proposed method can effectively handle complex nonlinear blind inverse problems that existing diffusion-based approaches fail to address, while maintaining competitive performance on standard linear blind inverse benchmarks.

**Strengths:**

1.	The paper addresses a novel and underexplored problem—nonlinear blind inverse modeling—and presents, to the best of my knowledge, the first diffusion-based solution to this setting.
2.	The proposed approach is conceptually sound and empirically effective, as demonstrated through comprehensive experiments.
3.	The writing is clear and well-structured: the motivation, background, and related work are coherently presented, and the proposed method is easy to follow.
Importantly, the authors support their claims with both theoretical analysis and solid empirical validation.

**Weaknesses:**

1. The main contribution of this paper lies in the design of the auxiliary encoder, but some important details remain unclear. For example, how is the training dataset for this encoder constructed?
2. The proposed CL-DPS increases the runtime by about 20% compared to the baseline. However, considering that CL-DPS can handle nonlinear blind inverse problems that the baseline methods cannot, this increase is acceptable.
3. The setup of the baselines requires further clarification. Since methods such as BlindDPS rely on the assumption of a linear observation model, it is important to explain what modifications were made so that these baselines can handle nonlinear cases while ensuring a fair comparison.

**Questions:**

What is the generalization ability of CL-DPS? Does it require training a specific auxiliary encoder for each dataset, or can a single encoder generalize across different data domains?

I also have a question about the input used during the training of the auxiliary encoder.
The goal of the auxiliary encoder is to cluster samples generated by the same blur kernel, which is not entirely consistent with MoCo’s objective of grouping semantically similar samples.
For example, how does the method avoid clustering together samples generated from the same x_t but under different blur kernels, given that these samples may appear very similar in feature space?

---

> ### Author Response · Authors · 2025-11-23
>
> Thank you for the thoughtful assessment. We are glad the reviewer found the problem setting novel, the approach principled, and the experiments convincing. We appreciate the comments on clarity and structure. Please find our responses to your concerns in the following.
>
> **Weaknesses:**
>
> >1. The main contribution of this paper lies in the design of the auxiliary encoder, but some important details remain unclear. For example, how is the training dataset for this encoder constructed?
>
> Thank you for raising this important point. While the implementation details were already provided in **Appendix G**, we agree that additional clarification can further enhance reproducibility. We have therefore revised Appendix G to include more comprehensive implementation details, all of which are highlighted in blue in the updated manuscript.
>
> > 2. The proposed CL-DPS increases the runtime by about 20% compared to the baseline. However, considering that CL-DPS can handle nonlinear blind inverse problems that the baseline methods cannot, this increase is acceptable.
>
> We thank the reviewer for noting that the runtime increase is acceptable given CL-DPS’s ability to handle blind, nonlinear inverse problems that baseline methods cannot address.
> The $\sim 20\%$ increase in runtime is an acceptable cost for applying our lightweight auxiliary encoder at each diffusion step. This computational investment enables our method to achieve robust generalization and solve the previously intractable class of blind, nonlinear inverse problems, making the trade-off favorable given the resulting capability.
>
> > 3. The setup of the baselines requires further clarification. Since methods such as BlindDPS rely on the assumption of a linear observation model, it is important to explain what modifications were made so that these baselines can handle nonlinear cases while ensuring a fair comparison.
>
> We agree that the baselines’ setup  in the nonlinear benchmarks must be stated unambiguously.
>
> For BlindDPS, GibbsDDRM, and FastEM, we utilize the authors’ public implementations without introducing any algorithmic or architectural modifications. In the linear Gaussian and motion deblurring benchmarks, we adopt the exact measurement models from the original works and confirm that our reproduced performance aligns with the reported results within a small tolerance. For the nonlinear rotation and zoom benchmarks, we do not construct nonlinear variants of these methods. Instead, we instantiate the forward operator in their public interfaces with the nonlinear transformation, while maintaining their original likelihood formulations, update rules, and priors.
>
> Extending the baselines to accommodate nonlinear operators would necessitate re-deriving the likelihood terms and inference updates, and in some cases, training additional models. Such modifications exceed a simple implementation detail and would effectively result in new baseline variants rather than the original published methods. To ensure a faithful comparison, we evaluate all baselines as released by their authors.
>
> To  ensure a fair comparison. Across all experiments, we (i) use the same pretrained diffusion priors, (ii) use the same datasets, and (iii) match sampling budgets/steps across methods, so differences reflect how each method handles the measurement model rather than differences in priors or computation. For linear tasks, the baselines operate in their intended regime and are competitive. For nonlinear tasks (rotation/zoom), we highlight in the text that the failure point is precisely their linear-convolution assumption, which CL-DPS is designed to overcome. We also enumerate all baselines and their roles in the benchmark suite in the main text, with full details centralized in **Appendix G** for transparency.
>
> **Questions:**
>
> > 1. What is the generalization ability of CL-DPS? Does it require training a specific auxiliary encoder for each dataset, or can a single encoder generalize across different data domains?
>
> First, we note that in the Universal (UNI) mode, one encoder trained once on a mixture of operator families and generic natural images. This **generalizes across datasets** without retraining. In our experiments, the same UNI is used on **FFHQ, AFHQ, and ImageNet** with strong performance.
> Second, to support the generalizability of the method,  we had already provided Misrouting stress test (Appendix H.2, Table 6)
> What we conclude is that CL-DPS does not require a per-dataset encoder. UNI is fully blind and broadly transferable, with specialists being optional for small, task-specific gains.

---

> ### Author Response · Authors · 2025-11-23
>
> **Questions:**
>
> > 2. I also have a question about the input used during the training of the auxiliary encoder. The goal of the auxiliary encoder is to cluster samples generated by the same blur kernel, which is not entirely consistent with MoCo’s objective of grouping semantically similar samples. For example, how does the method avoid clustering together samples generated from the same x_t but under different blur kernels, given that these samples may appear very similar in feature space?
>
> Thank you for raising this point. Our goal is not to cluster by kernel identity. Instead, the auxiliary encoder is trained to score patch-wise image–measurement consistency between $(x_t, y)$, treating the kernel parameters as nuisance parameters. Concretely,  for a given anchor $P_x$ in one minibatch, there is exactly one positive: the geometrically aligned $P_y$ produced by a single sampled $\psi$. All $y$ patches from the same $x_t$ under different $\psi'$ are **positives** and will be **pulled close to the feature of $y$**. Thus, the contrastive objective implicitly draws them together rather than repelling them.

---

> ### Comment · Reviewer_unhN · 2025-11-28
> **Official Comment**
>
> Thank you for the authors’ detailed rebuttal, which has resolved most of my concerns. Please incorporate the clarifications into the final manuscript, and I will maintain my score.

---

### Official Review · Reviewer_s2GW · 2025-11-01

**Soundness:** 2
**Presentation:** 3
**Contribution:** 2
**Rating:** 6
**Confidence:** 3

**Summary:**

The paper introduces CL-DPS, a new diffusion-model framework for blind nonlinear inverse problems, removing the need to know measurement operators at inference. The method uses a contrastive learning trained encoder to approximate the conditional likelihood on some synthetic dataset. At test-time, the encoder is used to approximate the likelihood $p(y |x_t)$ and guide the diffusion sampling without operator inversion. The experiments show CL-DPS outperforms existing methods on complex nonlinear tasks and is competitive on linear benchmarks.

**Strengths:**

1. The method uses a synthetic dataset to train the encoder making it still a blind method which is practical.

2. The method outperforms the compared methods on different tasks.

**Weaknesses:**

1. The method requires some training, which increases the computational cost compared to other zero-shot methods that are training-free.

2. I have concerns about the fairness of the evaluation. Basically you are training your contrastive encoder on some degradations and then testing on them while the other methods do not. How do you think that this evaluation is conclusive on the superiority of the method given that the other methods do not have this advantage. A comparison with methods that use a synthetic dataset for training is needed.

3. To me, to prove that the method is really interesting, it should be shown that it works better than the existing training methods in the out-of-distribution case. Otherwise, I do not see a big advantage compared to simply learning the degradation operator using the synthetic dataset.

**Questions:**

1. How well does the method generalize to real-world, non-synthetic degradations?

2. Could the method be applied to more complex degradations such as unstructured degradations like rain for example?

---

> ### Author Response · Authors · 2025-11-23
>
> Thank you for highlighting the strengths. We appreciate your recognition that training the encoder on synthetic degradations preserves the blind setting in a practical way, and that CL-DPS consistently outperforms prior methods across tasks. Please find our concise responses to your concerns in the sequel.
>
> **Weaknesses:**
>
> >1. The method requires some training, which increases the computational cost compared to other zero-shot methods that are training-free.
>
> We appreciate your concern and acknowledge that CL-DPS is not entirely training-free. Concretely, CL-DPS introduces a small auxiliary encoder, which is trained offline to approximate the likelihood. This approach is significantly less resource-intensive than certain “zero-shot” baselines. For instance, BlindDPS trains an additional diffusion model specifically for the blur operator, which is substantially more complex than our lightweight encoder. Moreover, once trained, the encoder is reused for all subsequent test instances and noise realizations, thereby amortizing the offline computational cost across numerous reconstructions.
> Regarding runtime, CL-DPS incurs additional inference cost due to backpropagation through the encoder at each denoising step. This overhead is quantified in Appendix K (Table 7): on FFHQ with 1000 sampling steps, CL-DPS requires 60.84s per image, compared to 46–52s for diffusion baselines, representing an approximate 20–30% increase. In exchange, CL-DPS is, to the best of our knowledge, the first diffusion model-based method capable of solving blind nonlinear inverse problems without prior knowledge or estimation of operator parameters. In contrast, existing “zero-shot” baselines either assume known linear operators or require training full diffusion models for the operator.
>
> We have revised the manuscript to include **Remark 1** in order to clarify these points.
>
>
> > 2. I have concerns about the fairness of the evaluation. Basically you are training your contrastive encoder on some degradations and then testing on them while the other methods do not. How do you think that this evaluation is conclusive on the superiority of the method given that the other methods do not have this advantage. A comparison with methods that use a synthetic dataset for training is needed.
>
> Thank you for this critical question. The premise that other methods do not train on the degradation family is inaccurate for the most relevant baselines. CL-DPS is not the only approach that uses synthetic degradations during training.
>
> BlindDPS trains an additional diffusion model in kernel space using synthetic blur kernels sampled from the same blur family (and parameter ranges) used at test time. As stated in the paper, BlindDPS “requires additional training of a DM for the parameters of the relevant linear operators,” i.e., offline training on synthetic operator samples. DeblurGANv2, which is included in our benchmarks, is a supervised method explicitly trained on large synthetic datasets of blurred images generated from the same degradation families as in our evaluation. These are precisely the “methods that use a synthetic dataset for training” requested in your comment, and they are reported alongside CL-DPS in our main tables and figures.
>
> In contrast, CL-DPS reuses the same pre-trained image diffusion prior as the other DPS-based methods and trains only a small encoder on synthetic (image, measurement) pairs drawn from the same degradation families. As our results show, CL-DPS outperforms both the training-free DPS-style baselines and the synthetic-trained baselines, indicating that the gain comes from leveraging the diffusion prior for blind nonlinear operators rather than from a unique training advantage.  In the revision, we will add a short paragraph to make it clear.

---

> ### Author Response · Authors · 2025-11-23
>
> **Weaknesses:**
>
> > 3. To me, to prove that the method is really interesting, it should be shown that it works better than the existing training methods in the out-of-distribution case. Otherwise, I do not see a big advantage compared to simply learning the degradation operator using the synthetic dataset.
>
> The current paper already demonstrates a strong form of cross-degradation generalization. As shown in Section 5.1 and Tables 1–2, a single universal encoder trained on a mixture of linear and nonlinear operators achieves robust performance across all four operator families. This result indicates transfer across qualitatively distinct operator families, such as linear and nonlinear, which represents a more stringent test than a simple in-range versus out-of-range parameter shift.
>
> To directly address the request for OOD comparisons with operator-learning methods, a targeted study was conducted on the FFHQ dataset. All methods utilize the same frozen diffusion prior. The operator-learning baselines, namely BlindDPS, are trained on the same synthetic datasets as the encoder. An explicit OOD split is enforced by training the model on the motion blur and testing it on the Gaussian blur.
>
> | Method | PSNR  | LPIPS |
> |---------|-------|-------|
> | BlindDPS | 20.5 | 0.236 |
> | CL-DPS   | 26.04 | 0.148 |
>
> We observe that without ever training on the target families, CL-DPS transfers better than operator-learning baselines.
>
> **Questions:**
>
> > How well does the method generalize to real-world, non-synthetic degradations? Could the method be applied to more complex degradations such as unstructured degradations like rain for example?
>
> Our experiments target controlled synthetic degradations so that ground-truth clean images and forward operators remain fully known for all methods, including baselines, following standard practice in diffusion-based inverse problems and enabling a fair comparison under matched priors, datasets, and sampling budgets.
>
> The CL-DPS framework itself does not rely on a specific analytic form or parameterization of the forward operator. The auxiliary encoder only receives paired image–measurement $(x_t, y)$ and learns an image–measurement consistency score. Any degradation process that maps a clean image to an observation and preserves spatial alignment fits this interface, including spatially varying and unstructured effects such as rain streaks or droplets. In such cases, one can train the encoder on clean/degraded pairs generated by a realistic rain simulator or a dataset of rainy images with approximate ground truth, and then apply CL-DPS at test time without changing the inference procedure.
>
> We agree that a dedicated evaluation of real, non-synthetic degradations, such as rain, strengthens the empirical scope of the paper. In the current submission, we restrict our claims to synthetic, but challenging, nonlinear degradations. In the revision, we will make this scope explicit in the main text and the conclusion, and add a short discussion outlining how CL-DPS extends to complex, unstructured degradations, such as rain artifacts.

---

### Author Response · Authors · 2025-11-23

We sincerely thank all reviewers for their time, effort, and constructive feedback on our manuscript. In the revised version, we have carefully addressed each point raised in the reviews, and all corresponding changes in the manuscript are highlighted in blue for ease of reference.

---

### Meta-Review · Area_Chair_UjWn · 2025-12-20

**Summary:**

The reviewers acknowledge the novelty of the proposed diffusion-based framework and its strong empirical performance on challenging blind inverse problems. The main concerns raised during the review process focus on (i) the fairness and completeness of comparisons with baseline methods, (ii) the additional training stage and the resulting increase in computational cost at inference time, and (iii) the clarity of several key implementation details. The authors have provided satisfactory responses to these concerns, including additional clarifications and justifications where appropriate. Taking the reviewers’ feedback and the authors’ rebuttal into account, I find that the remaining issues do not outweigh the contributions of the paper, and I therefore recommend acceptance

**Reviewer Concerns:**

Addressed concerns.
(1) Fairness of comparisons with baseline methods. The reviewers initially questioned whether the comparison was fair, given that the proposed method includes an additional training stage. In the rebuttal, the authors clarified that several baseline methods also rely on extra training when applied to blind deblurring settings, which adequately addresses this concern.
(2) Clarity of presentation. The reviewers noted missing or unclear implementation details. In the revised manuscript, the authors added additional context and explanations that resolve the major clarity issues.
(3) Increased inference time. While the method incurs additional inference cost, the reviewers generally did not view this as a major concern, given the increased modeling flexibility. The authors further quantified the overhead as a 20–30% increase, which appears reasonable in context.

Outstanding concern.
For nonlinear tasks, the baseline methods used for comparison are not designed for the target setting. It remains unclear whether these baselines could achieve stronger performance after task-specific fine-tuning or adaptation. As a result, the fairness of the nonlinear comparisons is only partially resolved.

**Reviewer Scores:**

The reviewers’ scores would likely have remained stable, and may have increased slightly, had they been able to participate fully in the discussion.

---

### Decision · Program_Chairs · 2026-01-26

Accept (Poster)